# MCBiF: Measuring Topological Autocorrelation in Multiscale Clusterings via 2-Parameter Persistent Homology

**Juni Schindler**
Department of Mathematics, Imperial College London, UK
Department of Mathematical Modeling and Machine Learning, University of Zurich, Switzerland
`juni.schindler@uzh.ch`

**Mauricio Barahona**
Department of Mathematics, Imperial College London, UK
`m.barahona@imperial.ac.uk`

## ABSTRACT

Datasets often possess an intrinsic multiscale structure with meaningful descriptions at different levels of coarseness. Such datasets are naturally described as multi-resolution clusterings, i.e., not necessarily hierarchical sequences of partitions across scales. To analyse and compare such sequences, we use tools from topological data analysis and define the Multiscale Clustering Bifiltration (MCBiF), a 2-parameter filtration of abstract simplicial complexes that encodes cluster intersection patterns across scales. The MCBiF is a complete invariant of (non-hierarchical) sequences of partitions and can be interpreted as a higher-order extension of Sankey diagrams, which reduce to dendrograms for hierarchical sequences. We show that the multiparameter persistent homology (MPH) of the MCBiF yields a finitely presented and block decomposable module, and its stable Hilbert functions characterise the topological autocorrelation of the sequence of partitions. In particular, at dimension zero, the MPH captures violations of the refinement order of partitions, whereas at dimension one, the MPH captures higher-order inconsistencies between clusters across scales. We then demonstrate through experiments the use of MCBiF Hilbert functions as interpretable topological feature maps for downstream machine learning tasks, and show that MCBiF feature maps outperform both baseline features and representation learning methods on regression and classification tasks for non-hierarchical sequences of partitions. We also showcase an application of MCBiF to real-world data of non-hierarchical wild mice social grouping patterns across time.

## 1 INTRODUCTION

In many applications, datasets possess an intrinsic multiscale structure, whereby meaningful descriptions exist at different resolutions or levels of coarseness. Think, for instance, of the multi-resolution structure in commuter mobility patterns (Alessandretti et al., 2020; Schindler et al., 2023), communities in social networks (Beguerisse-Díaz et al., 2017) and thematic groups of documents (Blei et al., 2003; Grootendorst, 2022); the subgroupings in single-cell data (Hoekzema et al., 2022) or phylogenetic trees (Chan et al., 2013); and the functional substructures in proteins (Delvenne et al., 2010; Delmotte et al., 2011). In such cases, the natural description of the data goes beyond a single clustering and consists of a multiscale sequence of partitions parametrised by a scale parameter $t$. Traditionally, multiscale descriptions have emerged from hierarchical clustering, with $t$ as the depth of the dendrogram (Carlsson & Mémoli, 2010; Murtagh & Contreras, 2012). However, in many real-world applications, the data structure is multiscale yet *non-hierarchical*. For example in temporal clustering, where $t$ corresponds to physical time (Rosvall & Bergstrom, 2010; Liechti & Bonhoeffer, 2020; Bovet et al., 2022); in topic modelling and document classification, where $t$ captures the coarseness of the topic groupings (Altuncu et al., 2019; Fukuyama et al., 2023; Liu et al., 2025); or

in clustering methods that exploit a diffusion on the data geometry, where $t$ is the increasing time horizon of the diffusion (Coifman et al., 2005; Azran & Ghahramani, 2006; Lambiotte et al., 2014).

A natural problem is then *how to analyse and compare non-hierarchical, multi-resolution sequences of partitions parametrised by the scale $t$*. Here we address this question from the perspective of topological data analysis (Carlsson & Zomorodian, 2009; Carlsson et al., 2009; Botnan & Lesnick, 2023) by introducing the *Multiscale Clustering Bifiltration* (MCbiF), a 2-parameter filtration of abstract simplicial complexes that encodes the patterns of cluster intersections across all scales.

**Problem definition.** A *partition* $\pi$ of set $X = \{x_1, x_2, ..., x_N\}$ is a collection of mutually exclusive subsets $C_i \subseteq X$ (or *clusters*) that cover $X$, i.e., $\pi = \{C_1, \ldots, C_c\}$ such that $X = \bigcup_{i=1}^{c} C_i$ and $C_i \bigcap C_j = \emptyset$, $\forall i \neq j$. We denote the cardinality as $|\pi| = c$, and $\pi_i$ for the $i$-th cluster $C_i$ of $\pi$. Let $\Pi_X$ denote the *space of partitions* of $X$. We write $\pi \leq \pi'$ if every cluster in $\pi$ is contained in a cluster of $\pi'$. This *refinement* relation constitutes a partial order and leads to the *partition lattice* $(\Pi_X, \leq)$ with lower bound $\hat{0} := \{\{x_1\}, \ldots, \{x_N\}\}$ and upper bound $\hat{1} := \{X\}$ (Birkhoff, 1967).

Here, we consider a *sequence of partitions* defined as $\theta : [t_1, \infty) \to \Pi_X$, $t \mapsto \theta(t) \in \Pi_X$, i.e., a piecewise-constant function that assigns a partition of $X$ to each $t$. The scale $t$ has $M$ *change points* $t_1 < t_2 < ... < t_M$, so that $\theta(t) = \theta(t_m)$ for $t \in [t_m, t_{m+1})$, $m = 1, \ldots, M-1$, and $\theta(t) = \theta(t_M)$ for $t \in [t_M, \infty)$. The sequence $\theta$ is *hierarchical* in $[s, t]$ if we have a strict sequence of refinements: either agglomerative ($\theta(r_1) \leq \theta(r_2), \forall r_1, r_2 \in [s, t]$ with $r_1 \leq r_2$) or divisive ($\theta(r_1) \geq \theta(r_2), \forall r_1, r_2 \in [s, t]$ with $r_1 \leq r_2$). We say that $\theta$ is *strictly hierarchical* if it is hierarchical in $[t_1, \infty)$. The sequence $\theta$ is called *coarse-graining* if $|\theta(s)| \geq |\theta(t)|$, $\forall s \leq t$.[1] Conversely, $\theta$ is called *fine-graining* if $|\theta(s)| \leq |\theta(t)|$, $\forall s \leq t$.

Our goal is to characterise and analyse sequences of partitions $\theta$, including non-hierarchical ones, in an integrated manner, taking account of memory effects across the scale $t$.

**Remark 1.** *Here, we are not concerned with the task of computing the multiscale clustering (i.e., the sequence of partitions $\theta$) from dataset $X$, for which several methods exist. Rather, we take $\theta$ as a given, and aim to analyse its structure.*

**Remark 2.** *This problem is distinct from consensus clustering, which aims to produce a summary partition by combining a set of partitions obtained, e.g., from different optimisations or clustering algorithms (Strehl & Ghosh, 2002; Vega-Pons & Ruiz-Shulcloper, 2011).*

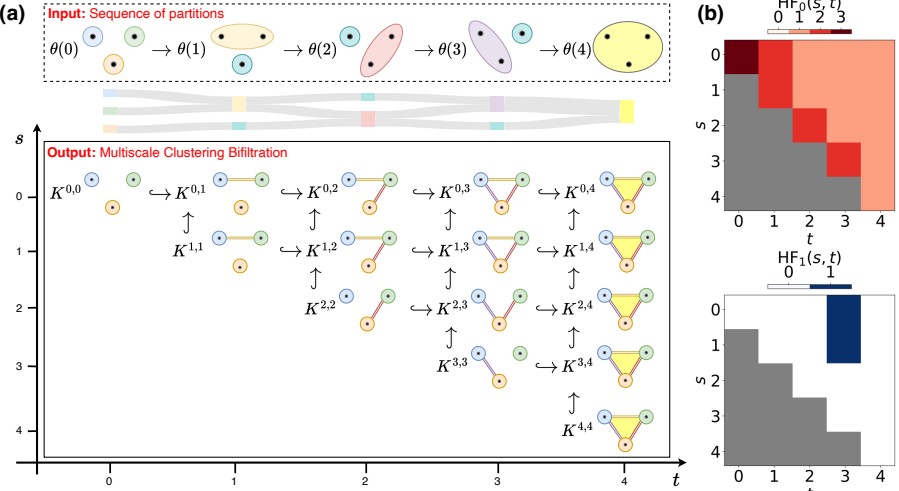

Figure 1: (a) The MCbiF encodes the structure of a non-hierarchical sequence of partitions $\theta$ as a bifiltration of abstract simplicial complexes $K^{s,t}$ (see Example 37). (b) The Hilbert functions $\mathrm{HF}_k(s, t)$ of the MCbiF are invariants that capture the topological autocorrelation of $\theta$: violations of the refinement order at dimension $k = 0$, and higher-order cluster inconsistencies at dimension $k = 1$. The $\mathrm{HF}_k(s, t)$ can be used as feature maps for downstream machine learning tasks.

---

[1]Coarse-graining is equivalent to non-decreasing mean cluster size (see Remark 23 in Appendix B.1).

**Contributions.** To address this problem, we define the MCbiF, a bifiltration of abstract simplicial complexes, which represents the clusters and their intersection patterns in the sequence $\theta$ for varying starting scale $s$ and lag $t - s$ (Fig. 1). The MCbiF is a complete invariant of $\theta$, and we use the machinery of multiparameter persistent homology (MPH) (Carlsson & Zomorodian, 2009; Carlsson et al., 2009; Botnan & Lesnick, 2023) to summarise its topological structure. We prove that the MCbiF leads to a block decomposable persistence module with stable Hilbert functions $\mathrm{HF}_k(s, t)$, and we show that these invariants characterise the *topological autocorrelation* of the sequence of partitions $\theta$ across scales. In particular, the $\mathrm{HF}_k(s, t)$ quantify the non-hierarchy in $\theta$ in complementary ways: at dimension $k = 0$, it detects the lack of a maximal partition in the subposet $\theta([s, t])$ with respect to refinement; at dimension $k = 1$, it quantifies higher-order inconsistencies of cluster assignments across scales. In contrast, baseline methods such as ultrametrics (Carlsson & Mémoli, 2010) or information-based measures (Meilă, 2003) are restricted to pairwise comparisons between elements or clusters, respectively; hence, they cannot detect higher-order cluster inconsistencies. Furthermore, we provide an equivalent nerve-based construction of the MCbiF that can be interpreted as a higher-order extension of the Sankey diagram of the sequence of partitions. In particular, for the hierarchical case, the 0-dimensional MCbiF Hilbert function can be obtained from the number of branches in the Sankey diagram, which reduces to a dendrogram. Finally, we show that $\mathrm{HF}_k(s, t)$ provide interpretable feature maps usable for downstream machine learning tasks. In our experiments, the MCbiF feature maps outperform both baseline features and representation learning methods on regression and classification tasks for non-hierarchical sequences of partitions. We also showcase an application of MCbiF to real-world data of non-hierarchical wild mice social grouping patterns across time (Bovet et al., 2022).

## 2 RELATED WORK

**Dendrograms and Ultrametrics.** A hierarchical, coarse-graining sequence $\theta$ with $\theta(t_1 = 0) = \hat{0}$ and $\theta(t_M) = \hat{1}$ is called an *agglomerative dendrogram*, and can be represented by an acyclic rooted merge tree (Jain et al., 1999; Carlsson & Mémoli, 2010). One can define an *ultrametric* $D_\theta$ from the first-merge times, which follows from the depth in the dendrogram. Carlsson & Mémoli (2010) showed there is a one-to-one correspondence between agglomerative dendrograms and ultrametrics that can be used to efficiently compare dendrograms using the Gromov-Hausdorff distance between the ultrametric spaces (Mémoli et al., 2023). When $\theta$ is non-hierarchical, however, first-merge times no longer define the sequence uniquely, as merged clusters can split again. Hence, $\theta$ cannot be represented by a tree, and $D_\theta$ does not fulfil the triangle inequality in general. As a result, ultrametrics cannot be used to analyse and compare non-hierarchical sequences of partitions (see Section 4).

**Pairwise Comparison of Partitions.** Different measures to compare pairs of partitions have been introduced in the literature. The *Adjusted Rand Index* (ARI) is the chance-corrected *Rand Index* that compares clusterings by counting elements that are assigned to the same or different clusters (Hubert & Arabie, 1985). Information-based measures compute the information gain and loss between two partitions using the *Conditional Entropy* (CE) or the *Variation of Information* (VI), which is a metric on $\Pi_X$ (Meilă, 2003; 2007) (see formulas in Appendix F). The *Maximum Overlap Distance* (MOD), which is also a metric on $\Pi_X$, measures the minimal classification error when one partition is assumed to be the correct one (Peixoto, 2021). A key limitation of all these measures is that they rely only on pairs of partitions and cannot capture higher-order cluster inconsistencies; yet extending them to more than two partitions is non-trivial. In consensus clustering, the average of the pairwise ARI, VI or MOD between the partitions in a set is used as a *consensus index* (Vinh & Epps, 2009; Vinh et al., 2010). However, these average measures are insensitive to ordering and so cannot capture memory effects in sequences of partitions.

## 3 THE MULTISCALE CLUSTERING BIFILTRATION (MCBIF)

The central object of our paper is a novel bifiltration of abstract simplicial complexes that encodes cluster intersection patterns in the sequence of partitions $\theta$ across the scale $t$. For background on simplicial complexes and bifiltrations, see Appendix E.2.

**Definition 3** (Multiscale Clustering Bifiltration). *Given a sequence $\theta : [t_1, \infty) \to \Pi_X$, we define the Multiscale Clustering Bifiltration (MCbiF), $\mathcal{M}$, as a bifiltration of abstract simplicial complexes:*

$$\mathcal{M} := (K^{s,t})_{t_1 \leq s \leq t} \quad where \quad K^{s,t} := \bigcup_{t_1 \leq s \leq r \leq t} \bigcup_{C \in \theta(r)} \Delta C. \tag{1}$$

In this construction, each cluster $C$ corresponds to a $(|C|-1)$-dimensional solid simplex $\Delta C := 2^C$, which, by definition, contains all its lower-dimensional simplices (Schindler & Barahona, 2025). This echoes natural concepts of data clustering viewed as information compression or lumping (Rosvall & Bergstrom, 2008; 2011; Lambiotte et al., 2014), and of clusters as equivalence classes (Brualdi, 2010). The MCbiF then aggregates all clusters (simplices) from partition $\theta(s)$ to $\theta(t)$ through the union operators, such that a $k$-simplex $\sigma = [x_1, \ldots, x_{k+1}] \in K^{s,t}$ consists of elements that are assigned to the same cluster (at least once) in the interval $[s,t]$, i.e., $x_1, \ldots, x_{k+1} \in C$ for some cluster $C \in \theta(r)$, $r \in [s,t]$. The bifiltration depends not only on the lag $|t-s|$ but also on the starting scale $s$, and captures the topological autocorrelation in the sequence of partitions, see Fig. 1. We first show that the MCbiF is indeed a well-defined bifiltration.

**Proposition 4.** *$\mathcal{M}$ is a multi-critical bifiltration uniquely defined by its values on the finite grid $P = \{(s,t) \in [t_1, \ldots, t_M] \times [t_1, \ldots, t_M] \mid s \leq t\}$ with partial order $(s,t) \leq (s',t')$ if $s \geq s', t \leq t'$.*

The proof is straightforward, see Appendix B.2. The MCbiF leads to a triangular commutative diagram where arrows indicate inclusion maps between abstract simplicial complexes (Fig. 1). The sequence of partitions $\theta(t)$ is encoded by the complexes $K^{t,t}$ on the diagonal of the diagram, hence the MCbiF is a complete invariant of $\theta$. Moving along horizontal arrows corresponds to fixing a starting scale $s$ and going forward in the sequence $\theta$, thus capturing coarse-graining. Moving along vertical arrows corresponds to fixing an end scale $t$ and going backwards in $\theta$, capturing fine-graining. By fixing $s := t_1$ (top row in the MCbiF diagram), we recover the 1-parameter Multiscale Clustering Filtration (MCF) defined by Schindler & Barahona (2025), see Remark 24.

Applying MPH to the bifiltration $\mathcal{M}$ at dimensions $k \leq \dim K$, for $K = K^{t_M, t_M}$, leads to a triangular diagram of simplicial complexes $H_k(K^{s,t})$ called persistence module (see Appendix E.2). We show in Proposition 27 in Appendix B.2 that the MCbiF persistence module is *pointwise finite-dimensional*, *finitely presentable* and *block-decomposable* (see Botnan & Lesnick (2023) for definitions). These strong algebraic properties are important because they guarantee algebraic stability of the MCbiF (Bjerkevik, 2021). In particular, the finite presentation property implies stability of the MCbiF Hilbert functions $\mathrm{HF}_k(s,t)$ (see Eq. 12 in Appendix E.2) with respect to small changes in the module (Oudot & Scoccola, 2024, Corollary 8.2.). This justifies the use of $\mathrm{HF}_k(s,t)$ as simple interpretable invariants for the topological autocorrelation captured by in the following.

### 3.1 MEASURING TOPOLOGICAL AUTOCORRELATION WITH MCBIF

We now show how the topological autocorrelation measured by the $\mathrm{HF}_k(s,t)$ of MCbiF can be used to detect cluster-assignment conflicts. We focus on dimensions $k = 0, 1$ in this paper, where $\mathrm{HF}_0(s,t)$ counts the number of connected components of $K^{s,t}$ and $\mathrm{HF}_1(s,t)$ is the number of 1-dimensional holes in $K^{s,t}$ (see Appendix E.2). We show below that the computation of these invariants reveals different aspects of the non-hierarchy in the sequence of partitions.

**Low-order Non-Hierarchy in Sequences of Partitions** An important aspect of hierarchy is the *nestedness* of the clusters in the sequence. We say that $\theta$ is *nested* in $[s,t]$ when $\forall r_1, r_2 \in [s,t]$, we have that $\forall C \in \theta(r_1), C' \in \theta(r_2)$, one of the sets $C \setminus C', C' \setminus C$ or $C \cap C'$ is empty. See Korte & Vygen (2012, Definition 2.12). We say that $\theta$ is *strictly nested* when $\theta$ is nested in $[t_1, \infty)$. It follows directly that a hierarchical sequence $\theta$ is always nested. However, the converse is not necessarily true, as illustrated by the nested, non-hierarchical sequence in Fig. 2b. To quantify this low-order non-hierarchy in sequence $\theta$, we can compute the invariant $\mathrm{HF}_0(s,t)$ with its associated notion of *0-conflicts* defined next. Recall that each partition $\theta(t)$ can be interpreted as an equivalence relation $\sim_t$ where $x \sim_t y$ if $\exists C \in \theta(t)$ such that $x, y \in C$ (Brualdi, 2010).

**Definition 5** (0-conflict and triangle 0-conflict). *a) We say that $\theta$ has a 0-conflict in $[s,t]$ if the subposet $\theta([s,t])$ has no maximum, i.e., $\nexists r \in [s,t]$ such that $\theta(r') \leq \theta(r)$, $\forall r' \in [s,t]$. b) We say that $\theta$ has a triangle 0-conflict in $[s,t]$ if $\exists x, y, z \in X$ such that $\exists r_1, r_2 \in [s,t]$: $x \sim_{r_1} y \sim_{r_2} z$ and $\nexists r \in [s,t]$: $x \sim_r y \sim_r z$.*

Next, we show that all triangle 0-conflicts are 0-conflicts. Moreover, all 0-conflicts break hierarchy, and triangle 0-conflicts additionally break nestedness.

**Proposition 6.** *(i) Every triangle 0-conflict is a 0-conflict, but the opposite is not true. (ii) If $\theta$ has a 0-conflict in $[s,t]$, then $\theta$ is non-hierarchical in $[s,t]$. (iii) If $\theta$ is either coarse- or fine-graining but non-hierarchical in $[s,t]$, then $\theta$ has a 0-conflict in $[s,t]$. (iv) If $\theta$ has a triangle 0-conflict in $[s,t]$, then $\theta$ is non-nested in $[s,t]$.*

See Appendix B.2.1 for the simple proof. Fig. 2b illustrates a 0-conflict that is not a triangle 0-conflict, and Fig. 2c shows a triangle 0-conflict. The following proposition develops a sharp upper bound for $\mathrm{HF}_0$ that can be used to capture 0-conflicts.

**Proposition 7.** *(i) $\mathrm{HF}_0(s,t) \leq \min_{r \in [s,t]} |\theta(r)|, \ \forall [s,t] \subseteq [t_1, \infty)$. (ii) $\mathrm{HF}_0(s,t) < \min_{r \in [s,t]} |\theta(r)|$ iff $\theta$ has a 0-conflict in $[s,t]$. (iii) $\mathrm{HF}_0(s,t) = |\theta(r)|$ for $r \in [s,t]$ iff $\theta(r)$ is the maximum of the subposet $\theta([s,t])$.*

See Appendix B.2.1 for the proof. Proposition 7 shows that $\mathrm{HF}_0$ measures low-order non-hierarchy in $\theta$ by capturing 0-conflicts. To quantify this, we introduce a global normalised measure for $\theta$.

**Definition 8** (Average 0-conflict). *Let $T := t_M + \frac{t_M - t_1}{M-1}$. The average 0-conflict is defined as:*

$$0 \leq \bar{c}_0(\theta) := 1 - \frac{2}{|T-t_1|^2} \int_{t_1}^{T} \int_{s}^{T} \frac{\mathrm{HF}_0(s,t)}{\min_{r \in [s,t]} \mathrm{HF}_0(r,r)} \mathrm{d}s \, \mathrm{d}t \leq 1. \tag{2}$$

Higher values of $\bar{c}_0(\theta)$ indicate a high level of 0-conflicts and increased low-order non-hierarchy, as shown by the next corollary.

**Corollary 9.** *(i) If $\theta$ is hierarchical in $[s,t]$, then $\mathrm{HF}_0(s,t) = \min(|\theta(s)|, |\theta(t)|)$. As a special case, this implies $\mathrm{HF}_0(t,t) = |\theta(t)|, \forall t \geq t_1$. (ii) $\bar{c}_0(\theta) > 0$ iff $\theta$ has a 0-conflict. (iii) Let $\theta$ be either coarse- or fine-graining. Then, $\bar{c}_0(\theta) = 0$ iff $\theta$ is strictly hierarchical.*

See Appendix B.2.1 for the proof. Furthermore, we can detect triangle 0-conflicts by analysing the graph-theoretic properties of the MCbiF 1-skeleton $K_1^{s,t}$. Proposition 30 in Appendix B.2.1 shows that a *clustering coefficient* $C(K_1^{s,t}) < 1$ indicates the presence of a triangle 0-conflict.

**Higher-order Inconsistencies between Clusters in Sequences of Partitions** Measuring 0-conflicts in $\theta$ is only one way of capturing non-hierarchy. An additional phenomenon that can arise in non-hierarchical sequences is higher-order inconsistencies of cluster assignments across scales. These are already captured by the 1-dimensional homology groups (Schindler & Barahona, 2025) and the associated notion of 1-conflict, which we define next. Recall the definition of 1-cycles $Z_1(K^{s,t})$ and non-bounding cycles $H_1(K^{s,t})$, see Eq.14 in Appendix E.3.

**Definition 10** (1-conflict). *We say that $\theta$ has a 1-conflict in $[s,t]$ if $\exists \, x_1, \ldots, x_n \in X$ such that the 1-cycle $z = [x_1, x_2] + \cdots + [x_{n-1}, x_n] + [x_n, x_1] \in Z_1(K^{s,t})$ is non-bounding; in other words, $[z] \in H_1(K^{s,t})$ with $[z] \neq 0$.*

The number of distinct 1-conflicts in $[s,t]$ (up to equivalence of the homology classes) is given by $\mathrm{HF}_1(s,t)$. We first show that 1-conflicts also lead to triangle 0-conflicts and thus break hierarchy and nestedness of $\theta$.

**Proposition 11.** *(i) $\mathrm{HF}_1(s,t) \geq 1$ iff $\theta$ has a 1-conflict in $[s,t]$. (ii) If $\theta$ has a 1-conflict in $[s,t]$, then it also has a triangle 0-conflict. (iii) If $\theta$ is hierarchical in $[s,t]$, then $\mathrm{HF}_1(s,t) = 0$.*

See Appendix B.2.1 for a proof. Proposition 11 shows that a 1-conflict is a special kind of triangle 0-conflict arising from higher-order cluster inconsistencies across scales. This is illustrated in Fig. 1 and Examples 37–38 in Appendix C, which present sequences of partitions where different 1-conflicts emerge across scales. Note that Proposition 11 (iii) shows that the MCbiF has a trivial 1-dimensional MPH if $\theta$ is strictly hierarchical.

**Remark 12.** *The presence of a 1-conflict in $[s,t]$ signals the fact that assigning all the elements involved in the conflict to a shared cluster would increase the consistency of the sequence $\theta$. Hence, when a 1-conflict gets resolved (e.g., the corresponding homology generator dies in the MPH at $(s, t')$, $t < t'$), then $\theta(t')$ is called a conflict-resolving partition (Schindler & Barahona, 2025). This is illustrated in Example 37, where a 1-conflict gets resolved by $\theta(4) = \hat{1}$, and Example 38, where three different 1-conflicts get resolved one by one by partitions $\theta(7)$, $\theta(8)$ and $\theta(9)$.*

To quantify the presence of 1-conflicts in $\theta$, we introduce an unnormalised global measure.

**Definition 13** (Average 1-conflict). *Let $T$ be as in Definition 8. The average 1-conflict is defined as:*

$$0 \leq \bar{c}_1(\theta) := \frac{2}{|T - t_1|^2} \int_{t_1}^{T} \int_{s}^{T} \mathrm{HF}_1(s,t)\mathrm{d}s\,\mathrm{d}t. \tag{3}$$

**Corollary 14.** $\bar{c}_1(\theta) > 0$ *iff $\theta$ has a 1-conflict. In particular, if $\theta$ is strictly nested, then $\bar{c}_1(\theta) = 0$.*

Figure 4 in Appendix A summarises all of our theoretical results and their relationships.

## 3.2 MCBIF AS A HIGHER-ORDER SANKEY DIAGRAM

Consider the definition of the Sankey diagram of the sequence of partitions provided in Section E.1, and its associated representation as an $M$-layered graph with vertices $V_m$ at each layer representing the clusters of $\theta(t_m)$, see Eq. 9. Let us define the disjoint union $A(\ell, m) := V_\ell \uplus ... \uplus V_m$, $1 \leq \ell \leq m$, which assigns an index to each cluster in $\theta(t)$ for $t \in [t_\ell, t_m]$. Furthermore, recall that $\theta(t)_i$ denotes the $i$-th cluster $C_i$ of $\theta(t)$. The nerve-based MCbiF can then be defined as follows.

**Definition 15** (Nerve-based MCbiF). *Let $s \in [t_\ell, t_{\ell+1})$, $\ell = 1, ..., M-1$, and $t \in [t_m, t_{m+1})$, $m = \ell, ..., M-1$ or $t \geq t_m$ for $m = M$. We define the nerve-based MCbiF as*

$$\tilde{\mathcal{M}} := (\tilde{K}^{s,t})_{t_1 \leq s \leq t}, \quad where \quad \tilde{K}^{s,t} := \{\sigma \subseteq A(\ell, m) : \bigcap_{(n,i) \in \sigma} \theta(t_n)_i \neq \emptyset\}. \tag{4}$$

The nerve-based MCbiF $\tilde{\mathcal{M}}$ is a 1-critical bifiltration with simplices representing clusters and their intersections, in contrast to the original MCbiF $\mathcal{M}$ (Eq. 1) in which simplices represent elements in $X$ and their equivalence relations. Despite these different perspectives, Proposition 34 in Appendix B.2.2 shows that $\tilde{\mathcal{M}}$ and $\mathcal{M}$ lead to the same MPH and can be considered as equivalent. However, the dimensionality of $\mathcal{M}$ and $\tilde{\mathcal{M}}$ can differ, as shown in the following proposition.

**Proposition 16.** *(i) $\dim K^{s,t} = \max_{s \leq r \leq t} \max_{C \in \theta(r)} |C| - 1$, $\forall t_1 \leq s \leq t$. (ii) $\dim \tilde{K}^{t_m, t_{m+n}} = n$, $\forall 1 \leq m \leq M, 0 \leq n \leq M - m$.*

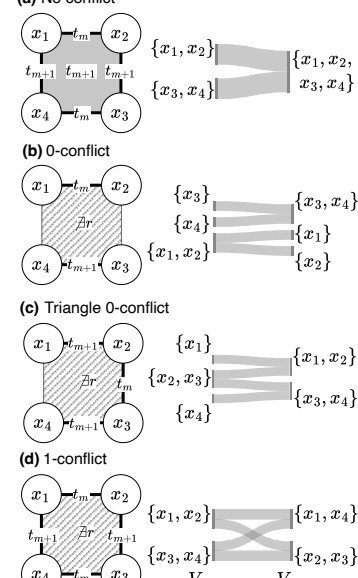

See proof in Appendix B.2.2. The nerve-based MCbiF is therefore computationally advantageous when the number of scales $M$ is smaller than the size of the largest cluster, $M < \max_{C \in \theta([t_1, \infty))} |C|$. The nerve-based $\tilde{\mathcal{M}}$ can be interpreted as a higher-order extension of the Sankey diagram $S(\theta)$ (Eq. 9). Yet, unlike $S(\theta)$, which only records pairwise intersections between clusters in consecutive partitions of $\theta$, $\tilde{\mathcal{M}}$ also accounts for higher-order intersections between clusters in sub-sequences of $\theta$.

**Proposition 17.** *The Sankey diagram graph $S(\theta)$ is a strict 1-dimensional subcomplex of $\tilde{K} := \tilde{K}^{t_1, t_M}$. In particular, $V_m = \tilde{K}^{t_m, t_m}$ and $E_m = \tilde{K}^{t_m, t_{m+1}}$, $\forall m = 1, \ldots, M-1$. Hence, we can retrieve $S(\theta)$ from the zigzag filtration*

$$\ldots \hookleftarrow \tilde{K}^{t_m, t_m} \hookrightarrow \tilde{K}^{t_m, t_{m+1}} \hookleftarrow \tilde{K}^{t_{m+1}, t_{m+1}} \hookrightarrow \ldots, \tag{5}$$

*which is a subfiltration of the nerve-based MCbiF.*

See proof in Appendix B.2.2. For details on zigzag persistence, see Carlsson & de Silva (2010) and Appendix E.4. Next, we characterise the 0- and 1-conflicts that can arise in a single layer $E_m$ of the Sankey diagram.

Figure 2: Characterisation of conflicts in a single-layer Sankey diagram, following Proposition 18.

**Proposition 18.** *(i) There is a 0-conflict in $[t_m, t_{m+1}]$ iff $\exists u \in V_m$ and $v \in V_{m+1}$ with $\deg(u) \geq 2$ and $\deg(v) \geq 2$, where $\deg$ denotes the node degree in the bipartite graph $(V_m \uplus V_{m+1}, E_m)$ associated with the Sankey diagram $S(\theta)$. (ii) There is a triangle 0-conflict in $[t_m, t_{m+1}]$ iff there is a path of length at least 3 in $E_m$. (iii) There is 1-conflict in $[t_m, t_{m+1}]$ iff there is a cycle in $E_m$.*

See Appendix B.2.2 for a proof and Fig. 2 for an illustration. Importantly, a cycle in $E_m$ leads to a crossing in $E_m$ that cannot be undone ( Fig. 2d). Hence, Proposition 18 (iii) implies that the sum of the elements of the superdiagonal of $\mathrm{HF}_1$ provides a lower bound for the minimal crossing number of the Sankey diagram, $\overline{\kappa}_\theta$ defined in Eq. 10.

**Corollary 19.** $\sum_{m=1}^{M-1} \mathrm{HF}_1(t_m, t_{m+1}) \leq \overline{\kappa}_\theta$.

**Remark 20.** *Note that 1-conflicts that arise across multiple partitions in the sequence (i.e., across multiple layers of the Sankey diagram) do not necessarily lead to crossings. See Fig. 1 for a Sankey diagram with no crossing despite the presence of a 1-conflict. However, we hypothesise that the full* $\mathrm{HF}_0$ *and* $\mathrm{HF}_1$ *feature maps capture more complicated crossings that arise in the Sankey layout across many layers. This insight is exploited in our computational tasks below.*

## 4 MATHEMATICAL LINKS OF MCBIF TO OTHER METHODS

**Ultrametrics.** Given a sequence $\theta$ with $\theta(t_1 = 0) = \hat{0}$ and $\theta(t_M) = \hat{1}$, let us define the matrix of *first-merge times* conditioned on the starting scale $s$:

$$D_{\theta,s}(x_i, x_j) := \min\{t \geq s \mid \exists\, C \in \theta(t) : x_i, x_j \in C\}. \tag{6}$$

When $s = 0$, this recovers the standard matrix of first-merge times $D_\theta := D_{\theta,0}$ in Section 2. If $\theta$ is hierarchical, i.e., an agglomerative dendrogram, then $D_\theta$ is an ultrametric, as it fulfils the *strong triangle inequality*: $D_\theta(x, z) \leq \max(D_\theta(x, y), D_\theta(x, z))\ \forall x, y, z \in X$. From Corollary 9 we have that the number of branches in the agglomerative dendrogram at level $t$, which is given by $|\theta(t)|$, is equal to $\mathrm{HF}_0(s, t)$ for any $s \leq t$. Hence, $\mathrm{HF}_0(s, t)$ contains the same information as the ultrametric in the hierarchical case, see Schindler & Barahona (2025) and Proposition 35. If, on the other hand, $\theta$ is non-hierarchical, triangle 0-conflicts can lead to violations of the (strong) triangle inequality.

**Proposition 21.** *The triplet $x, y, z \in X$ leads to a triangle 0-conflict in $[s, t]$ iff $x, y, z$ violate the strong triangle inequality for $D_{\theta,s}$, i.e., $D_{\theta,s}(x, z) > \max(D_{\theta,s}(x, y), D_{\theta,s}(y, z))$.*

See Appendix B.3 for a proof and Fig. 7a for an illustration. Proposition 21 shows that $\overline{c}_0(\theta)$ measures how much the ultrametric property of $D_\theta$ is violated. Recall that $D_{\theta,s}$ is a *dissimilarity measure* that can be used to define a filtration (Chazal et al., 2014). We show in Proposition 35 in Appendix B.3 that the 0-dimensional MPH of MCbiF corresponds to the 0-dimensional MPH of the *Merge-Rips bifiltration* constructed from $D_{\theta,s}$. In the hierarchical case, the Merge-Rips bifiltration has trivial 1-dimensional MPH, as $D_\theta$ fulfils the strong triangle inequality, and is thus equivalent to the MCbiF, whose 1-dimensional MPH is also trivial in the hierarchical case, see Proposition 11.

**Conditional Entropy.** The conditional entropy (CE) is only defined for pairs of partitions as the expected information of the conditional probability of $\theta(t)$ given $\theta(s)$, denoted $P_{t|s}$ (see Eq. 15 in Appendix F). For the case $M = 2$ (i.e., only two partitions in the sequence $\theta$), $\mathrm{HF}_0(t_1, t_2)$ follows directly from the spectral properties of the matrix $P_{t_2|t_1} P_{t_2|t_1}^T$ interpreted as an undirected graph.

**Proposition 22.** $\mathrm{HF}_0(t_1, t_2) = \dim(\ker L)$ *for graph Laplacian* $L = \mathrm{diag}(P_{t_2|t_1}\mathbf{1}) - P_{t_2|t_1} P_{t_2|t_1}^T$.

See Appendix B.3 for a proof. Note that $P_{t|s}$ only encodes the pairwise relationship between clusters, and does not capture higher-order cluster inconsistencies. In particular, CE cannot detect 1-conflicts arising across more than two scales, as seen in Example 39 in Appendix C.

## 5 EXPERIMENTS

### 5.1 REGRESSION TASK: MINIMAL CROSSING NUMBER OF SANKEY LAYOUT

Our first experiment considers a task of relevance in computer graphics and data visualisation: minimising the crossing number of Sankey diagram layouts (Zarate et al., 2018; Li et al., 2025). This minimisation is NP-complete. Here we use our MCbiF feature maps to predict the minimal crossing number $\overline{\kappa}_\theta$ (Eq. 11) of the Sankey diagram $S(\theta)$ of a given sequence of partitions $\theta$ (see Section E.1).

We build synthetic datasets by sampling randomly from $\Pi_N^M$, the space of coarse-graining sequences of partitions of $N$ elements with $M$ change points, see Definition 40 in Appendix D.1. Setting

$M = 20$, we generate two datasets of 20,000 random sequences $\theta \in \Pi_N^M$ for $N = 5$ and $N = 10$. For each $\theta$, we compute five feature maps: CE, ARI, MOD (see Section 2), and our MCbiF Hilbert functions ($\mathrm{HF}_0$ and $\mathrm{HF}_1$). To benchmark against representation learning, we also consider the (non-unique) *raw label encoding of $\theta$* given by the $N \times M$ matrix whose $m$-th column contains the labels of the clusters in $\theta(t_m)$ assigned to the elements in $X$, and the Sankey graphs $S(\theta)$ (Eq. 9). Our prediction target is $y = \overline{\kappa}_\theta$ (Eq. 11), the minimal crossing number of the layout of the Sankey diagram, as computed with the `OmicsSankey` algorithm (Li et al., 2025). See Section E.1 for details. We expect that predicting $y$ becomes harder for $N = 10$ because the increased complexity of $\Pi_M^N$ allows for more complicated crossings in the Sankey diagram.

As a preliminary assessment, we compute the Pearson correlation, $r$, between the crossing number $y$ and summary statistics of the five feature maps under investigation: the consensus indices $\overline{\mathrm{VI}}$, $\overline{\mathrm{ARI}}$ and $\overline{\mathrm{MOD}}$, and the MCbiF average measures $\bar{c}_0$ and $\bar{c}_1$. The correlation between the consensus indices and $y$ is low, higher for $\bar{c}_0$, and highest for $\bar{c}_1$ ($r = 0.47$ for $N = 5, 10$) (see Fig. 10 in Appendix D.1). This is consistent with our theoretical results in Section 3.2, which show the relation between the crossing number and $\mathrm{HF}_1$ (see Corollary 19).

Table 1: Regression task. Test $R^2$ score of LR, CNN, MLP and GCN models trained on different feature sets for $N = 5$ and $N = 10$. See Appendix D.1 for confidence intervals and train $R^2$ scores.

| $N$ | Method | Raw label encoding | Sankey graph | $\mathrm{HF}_0$ | $\mathrm{HF}_1$ | $\mathrm{HF}_0$ & $\mathrm{HF}_1$ | CE | ARI | MOD |
|---|---|---|---|---|---|---|---|---|---|
| 5 | LR | 0.078 | - | 0.147 | 0.486 | 0.539 | 0.392 | 0.166 | 0.413 |
| | CNN | 0.267 | - | 0.155 | 0.504 | **0.544** | 0.492 | 0.422 | 0.354 |
| | MLP | 0.104 | - | 0.150 | 0.491 | 0.541 | 0.409 | 0.214 | 0.351 |
| | GCN | - | 0.416 | - | - | - | - | - | - |
| 10 | LR | 0.038 | - | 0.214 | 0.448 | **0.516** | 0.457 | 0.246 | 0.345 |
| | CNN | 0.072 | - | 0.211 | 0.448 | 0.507 | 0.454 | 0.294 | 0.312 |
| | MLP | 0.036 | - | 0.212 | 0.450 | 0.514 | 0.458 | 0.256 | 0.246 |
| | GCN | - | 0.229 | - | - | - | - | - | - |

For the regression task of predicting $\overline{\kappa}_\theta$, we split each dataset into training (64%), validation (16%) and test (20%). For each feature map (or their combinations), we train three models: linear regression (LR), multilayer perceptron (MLP), and convolutional neural network (CNN). We use the mean-squared error (MSE) as our loss function and employ the validation set for hyperparameter tuning. See Appendix D.1 for details. We then evaluate model performance on the *unseen test data* using the coefficient of determination ($R^2$). As representation learning from the raw data, we also train models (LR, CNN, MLP) on the raw label encodings, and a graph convolutional network (GCN) on the Sankey graph $S(\theta)$ (Eq. 9). Table 1 shows that models trained on MCbiF feature maps outperform all representation learning models; even the GCN trained on Sankey graphs ($R^2 = 0.416$ for $N = 5$; $R^2 = 0.229$ for $N = 10$) has significantly lower performance ($p < 0.0001$, Wilcoxon test on residuals) than a simple LR model based on $\mathrm{HF}_1$ ($R^2 = 0.486$ for $N = 5$; $R^2 = 0.185$ for $N = 10$). This reflects the fact that the MCbiF is a higher-order extension of $S(\theta)$ that better captures the global properties of $\theta$ (see Proposition 17).

The MCbiF feature maps also outperform the baseline feature maps CE, ARI and MOD; the combined $\mathrm{HF}_0$ and $\mathrm{HF}_1$ features ($R^2 = 0.544$ for $N = 5$; $R^2 = 0.516$ for $N = 10$) have significantly better performance ($p < 0.0001$, Wilcoxon test) than the best baseline features CE ($R^2 = 0.492$ for $N = 5$; $R^2 = 0.458$ for $N = 10$). Variability analysis (bootstrapped 95% confidence intervals) shows that models trained on the combined $\mathrm{HF}_0$ and $\mathrm{HF}_1$ features outperform the other models (see Table 3 in Appendix D.1). Note that the strong performance of simple LR models underscores the interpretability of the MCbiF features, important for explainable AI (Adadi & Berrada, 2018).

## 5.2 CLASSIFICATION TASK: NON-ORDER-PRESERVING SEQUENCES OF PARTITIONS

In our second task, we classify whether a sequence of partitions is order-preserving, i.e., whether $\theta$ is compatible with a total ordering on the set $X$, see Definition 41 in Appendix D.2. This task is relevant in socio-economics (preference relations in utility theory in social sciences (Roberts, 2009)) and computer science (weak ordering and partition refinement algorithms (Habib et al., 1999)).

We carry out this classification task on synthetic data for which we have a ground truth. From the space of coarse-graining sequences of partitions $\Pi_N^M$ introduced in Definition 40, we generate a balanced dataset of 3,700 partitions $\theta \in \Pi_{500}^{30}$, half of which are order-preserving ($y = 0$) and the other half are non-order-preserving ($y = 1$). The loss of order-preservation is induced by introducing random swaps in the cluster assignments within layers. See Appendix D.2 for details. For each of the generated $\theta$ we compute CE, ARI, MOD, $HF_0$ and $HF_1$ using the computationally advantageous nerve-based MCbiF. We choose $N = 500$ and $M = 30$ to demonstrate the scalability of our method.

Whereas we find no significant difference between the baseline consensus indices of order-preserving ($y = 0$) and non-order-preserving ($y = 1$) sequences, we observe a significant increase of $\bar{c}_0$ and $\bar{c}_1$ for order-preserving sequences (Fig. 12 in Appendix D.2). For the classification task, we split our data into training (80%) and test

Table 2: Classification task. Test accuracy of logistic regression trained on different features. See Appendix D.2 for confidence intervals.

| Raw label encoding | $HF_0$ | $HF_1$ | CE | ARI | MOD |
|---|---|---|---|---|---|
| 0.53 | 0.56 | **0.97** | 0.50 | 0.49 | 0.46 |

(20%). For each feature map, we train a logistic regression on the training split and evaluate the accuracy on the test split, see Appendix D.2. We find that $HF_1$ predicts the label $y = \{0, 1\}$ encoding the (lack of) order-preservation with high accuracy (0.97). In contrast, CE, ARI, MOD and the raw label encoding of $\theta$ do not improve on a random classifier (Table 2). Our performance variability analysis (bootstrapped 95% confidence intervals) confirms that the model trained on $HF_1$ consistently outperforms all models (Table 5 in Appendix D.2). This demonstrates the high sensitivity of MCbiF to order-preservation in $\theta$, due to non-order-preserving sequences inducing 1-conflicts that are captured by $HF_1$.

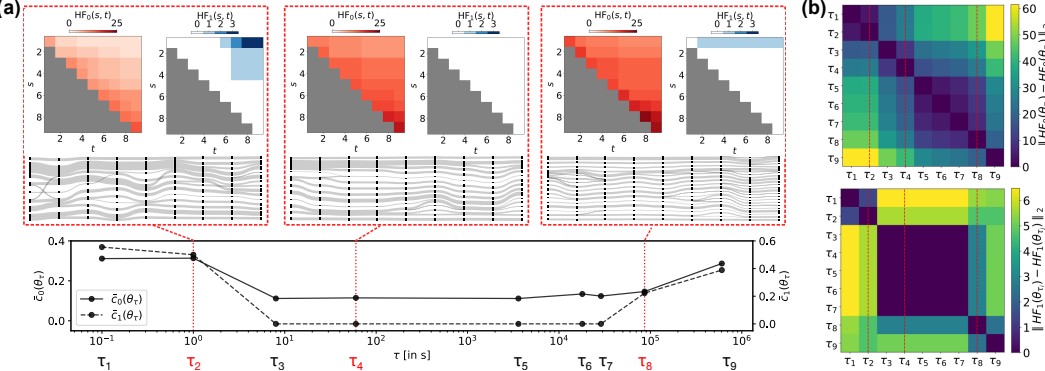

Figure 3: (a) Analysing non-hierarchical sequences of temporal partitions $\theta_{\tau_i}$ compiled from social interactions of a mice population over 9 weeks. Each $\theta_{\tau_i}$ is a sequence of social groupings $\theta_{\tau_i}(t)$ observed over week $t$. Different sequences of partitions are computed as a function of the parameter $\tau_i$. Sankey diagrams and MCbiF feature maps are shown for $\theta_{\tau_i}$ at $\tau_i$ ($i = 2, 4, 8$), identified as robust in Bovet et al. (2022). These three sequences exhibit different types of non-hierarchy, as shown by our topological feature maps and measures of average 0-conflict ($\bar{c}_0$) and average 1-conflict ($\bar{c}_1$). (b) The robust $\theta_{\tau_i}$ ($i = 2, 4, 8$) found in Bovet et al. (2022) correspond to distinct topological characteristics of the sequences of partitions, as captured by the block structure in the distance between MCbiF Hilbert functions. The Hilbert distances also capture increased time reversibility in $\theta_{\tau_4}$ and $\theta_{\tau_8}$ due to the larger stability of social groupings over time (see Fig. 14 in Appendix D.3).

## 5.3 Application to Real-World Temporal Data

In our final experiment, we apply MCbiF to temporal sequences of partitions computed from real-world contact data of free-ranging house mice. This data captures the changes in the social network structure of the rodents over time (Bovet et al., 2022). Each partition $\theta_\tau(t)$ describes mice social groupings for $N = 281$ individual mice at week $t \in [1, \ldots, 9]$ for the nine weeks in the period 28 February–1 May 2017, so that the sequence $\theta_\tau$ captures the fine-graining of social groups from winter to spring. A partition sequence is computed at temporal resolution $\tau > 0$, where the parameter $\tau$ modulates how fine the temporal community structure is (Fig. 13). See Bovet et al. (2022) and Ap-

pendix D.3 for details. We then compare temporal partition sequences $\theta_{\tau_i}$ for different resolutions $\tau_i$, $i = 1, \ldots, 9$, as given in Bovet et al. (2022). For each $\theta_{\tau_i}$, we obtain $\text{HF}_0$ and $\text{HF}_1$ using the nerve-based MCbiF, which induces a 50-fold reduction in computation time due to a much lower number of simplices (260 simplices for the nerve-based MCbiF *vs.* 116,700 for the original MCbiF).

Bovet et al. (2022) identified that the temporal resolutions $\tau_2 = 1\,\text{s}$, $\tau_4 = 60\,\text{s}$ and $\tau_8 = 24\,\text{h}$ lead to robust sequences of partitions. Using the Hilbert distance (i.e., the $L_2$-norm on the 0- and 1-dimensional MCbiF Hilbert functions), we find these temporal resolutions to be representative of three distinct temporal regimes characterised by different degrees of non-hierarchy, as measured by $\bar{c}_0$ and $\bar{c}_1$ (Fig. 3). In particular, high $\bar{c}_0$ indicates that mice tend to split off groups over time, and high $\bar{c}_1$ indicates that mice meet in overlapping subgroups but never jointly in one nest box. Note that $\theta_{\tau_2}$ has a strong non-hierarchical structure because the large-scale mice social clusters get disrupted in the transition to spring. In contrast, $\theta_{\tau_8}$ is more hierarchical as it captures the underlying stable social groups revealed by the higher temporal resolution. Yet, $\theta_{\tau_4}$ has the strongest hierarchy as indicated by a lower $\bar{c}_0$ and an absence of 1-conflicts ($\bar{c}_1 = 0$) and thus corresponds to a sweet spot in hierarchical organisation between low and high temporal resolutions. These findings also underpin the optimised Sankey diagrams in Figure 3a. While the Sankey diagram for $\theta_{\tau_4}$ can be drawn without any crossings, the optimal Sankey diagram for $\theta_{\tau_8}$ has a crossing from week 1 to week 2, which resembles the 'hourglass' 1-conflict (Fig. 2d). This is predicted by our MCbiF features: $\text{HF}_1(1, 2) = 1$ for $\theta_{\tau_8}$ implying one crossing between the first and second layer of the Sankey diagram that cannot be undone, as per Corollary 19.

## 6    CONCLUSION

We have introduced the MCbiF, a novel bifiltration that encodes the cluster intersection patterns in multiscale, non-hierarchical sequences of partitions. Its stable Hilbert functions $\text{HF}_k$ quantify the topological autocorrelation of the partition sequence $\theta$ and measure non-hierarchy in two complementary ways: at dimension $k = 0$ it captures the absence of a maximum with respect to the refinement order (0-conflicts), whereas at dimension $k = 1$ it captures the emergence of higher-order cluster inconsistencies (1-conflicts). This is summarised by the measures of average 0-conflict $\bar{c}_0(\theta)$ and average 1-conflict $\bar{c}_1(\theta)$, which are global, history-dependent and sensitive to the ordering of the partitions in $\theta$. The MCbiF extends the 1-parameter MCF defined by Schindler & Barahona (2025) to a 2-parameter filtration, leading to richer algebraic invariants that describe the full topological information in $\theta$. It is important to remark that the MCbiF is independent of the chosen clustering algorithm and can be applied to any (non-hierarchical) sequence of partitions $\theta$. We demonstrate with numerical experiments that the MCbiF Hilbert functions provide topological feature maps for downstream machine learning tasks, which are shown to outperform both baseline features and representation learning on the raw data for regression and classification tasks on non-hierarchical sequences of partitions. Moreover, the grounding of MCbiF features in algebraic topology enhances interpretability, a crucial attribute for explainable AI and applications to real-world data.

**Limitations and future work**    Our analysis of the MCbiF MPH is restricted to dimensions 0 and 1 for simplicity. However, the analysis of topological autocorrelation for higher dimensions would allow us to capture more complex higher-order cluster inconsistencies and could be the object of future research. Furthermore, we focused here on Hilbert functions as our topological invariants because of their computational efficiency and analytical simplicity, which facilitates our theoretical analysis. In future work, we plan to use richer feature maps by exploiting the block decomposition of the MCbiF persistence module, which leads to barcodes (Bjerkevik, 2021), or by using multiparameter persistence landscapes (Vipond, 2020). Another future direction is to use MCbiF to evaluate the consistency of assignments in consensus clustering (Strehl & Ghosh, 2002; Vega-Pons & Ruiz-Shulcloper, 2011). It can be shown that the values of the Hilbert function $\text{HF}_k(s, t)$ further away from the diagonal ($s = t$) are more robust to permuting the order of partitions in $\theta$ (see Proposition 36 in Appendix B.4). In particular, $\text{HF}_k(t_1, t_M)$ only depends on the set of distinct partitions in $\theta([t_1, \infty))$ and is independent of any permutation in their order. Hence, in future work, $\text{HF}_k(t_1, t_M)$ could be used as an overall measure of consistency in $\theta$ in the vein of consensus clustering. Finally, minimal cycle representatives of the MPH (Li et al., 2021) can be used to localise 1-conflicts in $\theta$, which is of interest to compute conflict-resolving partitions in consensus clustering, or to identify inconsistent assignments in temporal clustering (Liechti & Bonhoeffer, 2020).

## REPRODUCIBILITY STATEMENT

Detailed proofs of all theoretical results can be found in Appendix B and extensive documentation of our experiments in Appendix D. An implementation of our method and code to reproduce our numerical experiments is available at: `https://github.com/barahona-research-group/MCbiF`. The dataset studied in Section 5.3 is publicly available at: `https://dataverse.harvard.edu/file.xhtml?fileId=5657692`.

## ACKNOWLEDGMENTS

JS acknowledges support from the EPSRC (PhD studentship through the Department of Mathematics at Imperial College London). MB acknowledges support from EPSRC grant EP/N014529/1 supporting the EPSRC Centre for Mathematics of Precision Healthcare. This work benefited from discussions at the London-Oxford TDA seminar in November 2024. We thank Kevin Michalewicz, Asem Alaa, Christian Schindler and Jacopo Graldi for valuable discussions on computational aspects of this project, Alex Bovet for discussions on the temporal dataset studied in this paper, and Arne Wolf for discussions on the stability of multiparameter persistence modules. We also thank Heather Harrington for helpful discussions and for the opportunity to present work in progress at the Max Planck Institute of Molecular Cell Biology and Genetics Dresden (MPI-CBG) in July 2025.

## AUTHOR CONTRIBUTIONS

JS developed the mathematical theory, conducted all numerical experiments, and authored the initial draft of the paper. MB provided supervision, idea generation and guidance throughout the full research process. Both authors wrote and edited the final manuscript.

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

## APPENDICES

## A SUMMARY OF KEY THEORETICAL RESULTS

In Figure 4, we provide a summary of our theoretical results and their relationships.

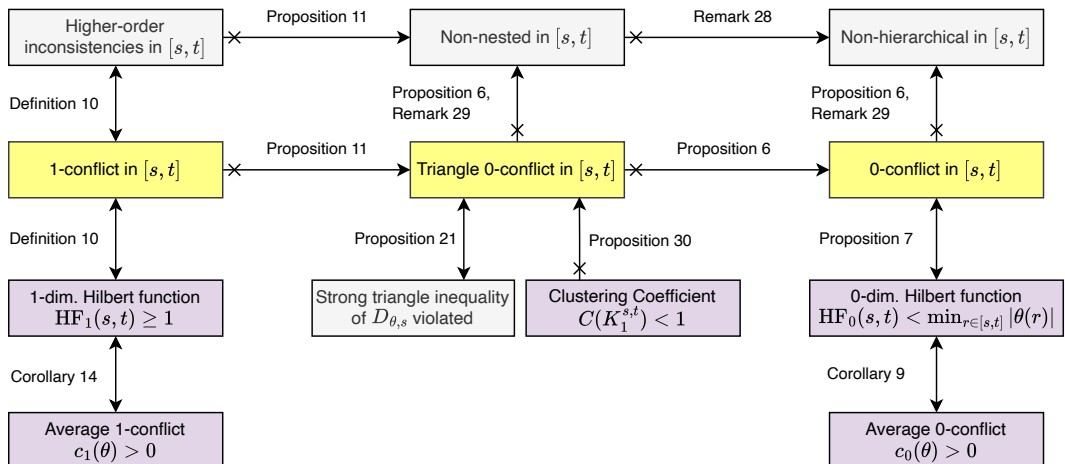

Figure 4: Summary of key theoretical results and their relationships. Double-headed arrows represent equivalences (iff), whereas single-headed arrows represent implications (if).

## B PROOFS OF THEORETICAL RESULTS AND ADDITIONAL DETAILS

### B.1 PROOFS AND DETAILS FOR SECTION 1

We first state a simple fact about coarse-graining sequences of partitions.

**Remark 23.** *Let $\theta(t)_i$ denote the $i$-th cluster $C_i$ of $\theta(t)$. It is a simple fact that $\theta$ is coarse-graining iff the mean cluster size is non-decreasing, i.e., $\frac{1}{|\theta(s)|} \sum_i^{|\theta(s)|} |\theta(s)_i| \leq \frac{1}{|\theta(t)|} \sum_j^{|\theta(t)|} |\theta(t)_j|$ for $s \leq t$. The proof follows directly from the fact that $\sum_i^{|\theta(s)|} |\theta(s)_i| = \sum_i^{|\theta(t)|} |\theta(t)_i| = N$.*

### B.2 PROOFS AND DETAILS FOR SECTION 3

We provide a proof for the multi-criticality of the MCbiF filtration stated in Proposition 4.

*Proof of Proposition 4.* The MCbiF is indeed a bifiltration because $K^{s,t} \subseteq K^{s',t'}$ if $s \geq s'$ and $t \leq t'$. See Fig. 5 for the triangular diagram of the MCbiF filtration, where arrows indicate inclusion maps. The MCbiF is uniquely defined by its values on the finite grid $[t_1, \ldots, t_M] \times [t_1, \ldots, t_M]$ because $\theta$ has change points $t_1 < \cdots < t_M$. It is a multi-critical bifiltration because for $x \in X$ we have $[x] \in K^{s,t}$ for all $s, t \in [t_1, \infty)^{\mathrm{op}} \times [t_1, \infty)$. In particular, $x \in K^{t_1,t_1}$ and $x \in K^{t_1+\delta,t_1+\delta}$ for $\delta > 0$ but $(t_1, t_1)$ and $(t_1 + \delta, t_1 + \delta)$ are incomparable in the poset $[t_1, \infty)^{\mathrm{op}} \times [t_1, \infty)$. □

**Remark 24.** *By fixing $s := t_1$ (i.e., the top row in the commutative MCbiF diagram), we recover the 1-parameter Multiscale Clustering Filtration (MCF) defined by Schindler & Barahona (2025). MCF was designed to quantify non-hierarchies in coarse-graining sequences of partitions and thus cannot capture fine-graining. For example, a large cluster $C \in \theta(s')$ prevents MCF from detecting cluster assignment conflicts between elements $x, y \in C$ for $t \geq s'$, see Section 3.1. In particular, the MCF is not a complete invariant of $\theta$. In contrast, MCbiF is a complete invariant of $\theta$ and encodes the full topological autocorrelation contained in $\theta$ by varying both the starting scale $s$ and the lag $t - s$.*

Next, we provide formal definitions for algebraic properties of persistence modules, see Botnan & Lesnick (2023) for details.

$$
\begin{array}{ccccc}
\cdots & & \cdots & & \cdots \\
\uparrow & & \uparrow & & \uparrow \\
K^{t,t} \lhook\joinrel\longrightarrow & K^{t,t'} \lhook\joinrel\longrightarrow & K^{t,t''} \lhook\joinrel\longrightarrow & \cdots \\
& \uparrow & & \uparrow & \\
& K^{t',t'} \lhook\joinrel\longrightarrow & K^{t',t''} \lhook\joinrel\longrightarrow & \cdots \\
& & \uparrow & \\
& & K^{t'',t''} \lhook\joinrel\longrightarrow & \cdots
\end{array}
$$

Figure 5: Triangular commutative diagram of the MCbiF for $t_1 \leq t \leq t' \leq t''$. The arrows indicate inclusion maps between simplicial complexes.

**Definition 25.** *For partially ordered sets $P_1, P_2$, we call an interval $I \subseteq P_1 \times P_2$ a block if it can be written as one of the following types:*

1. *Birth quadrant: $I = S_1 \times S_2$ for downsets $S_1 \subseteq P_1$ and $S_2 \subseteq P_2$.*

2. *Death quadrant: $I = S_1 \times S_2$ for upsets $S_1 \subseteq P_1$ and $S_2 \subseteq P_2$.*

3. *Vertical band: $I = S_1 \times P_2$ for an interval $S_1 \subseteq P_1$.*

4. *Horizontal band: $I = P_1 \times S_2$ for an interval $S_2 \subseteq P_2$.*

**Definition 26.** *Let **Vect** denote the category of $k$-vector spaces for a fixed field $k$. For a partially ordered set $P$, a $P$-indexed persistence module is a functor $F : P \mapsto$ **Vect**. We say that:*

a) *$F$ is called pointwise finite-dimensional if $\dim(F_a) < \infty$ for all $a \in P$.*

b) *$F$ is called finitely presented if there exists a morphism of free modules $\phi_1 : F_1 \to F_1$ such that $\operatorname{coker}(\phi_1) \cong F$ and $F_0$ and $F_1$ are finitely generated.*

c) *$F$ is called block-decomposable if it decomposes into blocks $F \bigoplus_{B \in \mathcal{B}(F)} k_B$ where $\mathcal{B}(F)$ is a multiset of blocks (see Definition 25) that depends on $F$.*

We can now provide the proof for Proposition 27, which shows that the MCbiF persistence module (see Fig. 6) is pointwise finite-dimensional, finitely presented and block-decomposable. The proof relies on the equivalent nerve-based construction of the MCbiF (see Proposition 34), and the exactness of the persistence module from which block-decomposability follows (Cochoy & Oudot, 2020).

**Proposition 27.** *For any $k \leq \dim K$, the MCbiF persistence module $H_k(K^{s,t})$ is pointwise finite-dimensional, finitely presented and block-decomposable.*

*Proof of Proposition 27.* The MCbiF module is pointwise finite-dimensional because the homology groups of finite simplicial complexes are finite. As the MCbiF is defined uniquely by its values on a finite grid (Proposition 4), its persistence module consists of finitely many vector spaces and finitely many linear maps between them, hence it is finitely presented.

To prove block-decomposability, we use the nerve-based MCbiF $(\tilde{K}^{s,t})_{t_1 \leq s \leq t}$, which leads to the same persistence module, see Proposition 34. As the module is uniquely defined by its values on a finite grid, we can use Theorem 9.6 by Cochoy & Oudot (2020) that implies block-decomposability if the persistence module is *exact*. Hence, it suffices to show that for all $t_1 \leq t \leq t' \leq t'' \leq t'''$ the diagram

$$
\begin{array}{ccc}
H_k(\tilde{K}^{t,t''}) & \longrightarrow & H_k(\tilde{K}^{t,t'''}) \\
\uparrow & & \uparrow \\
H_k(\tilde{K}^{t',t''}) & \longrightarrow & H_k(\tilde{K}^{t',t'''})
\end{array}
$$

induces an exact sequence:

$$
H_k(\tilde{K}^{t',t''}) \to H_k(\tilde{K}^{t,t''}) \oplus H_k(\tilde{K}^{t',t'''}) \to H_k(\tilde{K}^{t,t'''}) \tag{7}
$$

By construction of the MCbiF, $\tilde{K}^{t,t'''} = \tilde{K}^{t,t''} \cup \tilde{K}^{t',t'''}$. Furthermore, $\tilde{K}^{t,t''} = \tilde{K}^{t,t'} \cup \tilde{K}^{t',t''}$ and $\tilde{K}^{t',t'''} = \tilde{K}^{t',t''} \cup \tilde{K}^{t'',t'''}$. As $\theta$ is a piecewise-constant function with change points $t_1 < \cdots < t_M$, we can assume without loss of generality, $t = t_k$, $t' = t_\ell$, $t'' = t_m$, $t''' = t_n$ for change points $t_k < t_\ell < t_m < t_n$ of $\theta$ such that $A(k,\ell) \cap A(m,n) = \emptyset$. Hence, $\tilde{K}^{t,t'} \cap \tilde{K}^{t'',t'''} = \emptyset$ and $\tilde{K}^{t',t''} = \tilde{K}^{t,t''} \cap \tilde{K}^{t',t'''}$. This means that Eq. equation 7 is a Mayer-Vietoris sequence for all $k \geq 0$, implying exactness (Hatcher, 2002, p. 149) and proving the block decomposability (Cochoy & Oudot, 2020, Theorem 9.6). $\square$

$$
\begin{array}{ccccc}
\cdots & & \cdots & & \cdots \\
\uparrow & & \uparrow & & \uparrow \\
H_k(K^{t,t}) \rightarrow & H_k(K^{t,t'}) \longrightarrow & H_k(K^{t,t''}) \rightarrow & \cdots \\
& & \uparrow & & \uparrow \\
& & H_k(K^{t',t'}) \rightarrow & H_k(K^{t',t''}) \rightarrow & \cdots \\
& & & & \uparrow \\
& & & & H_k(K^{t'',t''}) \rightarrow \cdots
\end{array}
$$

Figure 6: Multiparameter persistence module of the MCbiF for $t_1 \leq t \leq t' \leq t''$. The arrows indicate linear maps between vector spaces.

### B.2.1 PROOFS AND DETAILS FOR SECTION 3.1

**Remark 28.** *It follows directly from the definitions that a hierarchical sequence $\theta$ is always nested. However, nestedness does not necessarily imply hierarchy, as illustrated by the example in Fig. 2b.*

We continue with the proof of Proposition 6 that relates 0-conflicts to hierarchy and triangle 0-conflicts to nestedness.

*Proof of Proposition 6.* (ii) If $\theta$ has a 0-conflict then $\exists r_1, r_2 \in [s,t]$ such that $\theta(r_1) \not\leq \theta(r_2)$ and $\theta(r_1) \not\geq \theta(r_2)$, otherwise $\theta([s,t])$ would have a maximum. Hence, $\theta$ is not hierarchical in $[s,t]$.

(iii) Let us first assume that $\theta$ is coarse-graining, i.e., $|\theta(t)| \leq |\theta(r)|$ for all $r \in [s,t]$. We show that no 0-conflict in $[s,t]$ implies that $\theta$ is hierarchical in $[s,t]$. Let $r_1, r_2 \in [s,t]$ with $r_1 \leq r_2$, then the subposet $\theta([r_1, r_2])$ has a maximum because of the absence of a 0-conflict, and the maximum is given by $\theta(r_2)$ due to coarse-graining. Hence, $\theta(r_1) \leq \theta(r_2)$. As $r_1, r_2$ were chosen arbitrarily, this implies that $\theta$ is hierarchical in $[s,t]$. The argument is analogous for the case that $\theta$ is fine-graining.

(iv) Let $x, y, z \in X$ be in a triangle 0-conflict. In particular, $x \sim_{r_1} y \sim_{r_2} z$ with $x \neq y$, $x \neq z$ and $y \neq z$. Hence, there are $C \in \theta(r_1)$ and $C' \in \theta(r_2)$ such that $x, y \in C$ and $y, z \in C'$, as well as $z \notin C$ and $x \notin C'$. This implies $\{x\} \in C \setminus C'$, $\{z\} \in C' \setminus C$ and $\{y\} \in C \cap C'$, showing that $C$ and $C'$ are non-nested. Hence, $\theta$ is non-nested in $[s,t]$.

(i) Moreover, $\nexists r \in [s,t]$ such that $x \sim_r \sim_r y$. In particular, $\nexists r \in [s,t]$ such that $\nexists C'' \in \theta(r)$ with $C \subseteq C''$ and $C \subseteq C'''$. Hence, $\nexists r \in [s,t]$ such that $\theta(r_1) \leq \theta(r)$ and $\theta(r_2) \leq \theta(r)$, implying that the subposet $\theta([s,t])$ has no maximum. This shows that every triangle 0-conflict is also a 0-conflict, proving statement (i). Note that the opposite is not true, as illustrated by the example in Fig. 2b. $\square$

**Remark 29.** *Non-nestedness and non-hierarchy do not imply the presence of a 0-conflict. To see this, consider the simple counter-example given by $\theta(0) = \{\{x,y\}, \{z\}\}$, $\theta(1) = \hat{1}$, $\theta(2) = \{\{x\}, \{y,z\}\}$, which is non-nested but the partition $\theta(1)$ is the maximum of the subposet $\theta([0,1,2])$. This illustrates the need for the additional assumption of coarse- or fine-graining of $\theta$ in Proposition 6 (iii) for the condition of no 0-conflict to imply hierarchy.*

We now provide a proof for Proposition 7 on properties of the 0-dimensional Hilbert function of the MCbiF.

*Proof of Proposition 7.* (i) $\mathrm{HF}_0(s,t)$ is equal to the number of connected components of $K^{s,t}$. Let $r' \in [s,t]$ such that $c = |\theta(r')| = \min_{r \in [s,t]} |\theta(r)|$. We can represent $\theta(r) = \{C_1, \ldots, C_c\}$ and

by construction $\Delta C \in K^{s,t}$ for all $C \in \theta(r)$. Hence, if two elements $x, y \in X$ are in the same cluster $C \in \theta(r)$ then $[x, y] \in K^{s,t}$ and the 0-simplices $[x], [y] \in K^{s,t}$ are in the same connected component. As $\theta(r)$ has $c$ mutually disjoint clusters, this means that there cannot be more than $c$ disconnected components in $K^{s,t}$ and $\mathrm{HF}_0(s, t) \leq c = |\theta(r')|$. As $r' \in [s, t]$ was chosen arbitrarily, this implies $\mathrm{HF}_0(s, t) \leq \min_{r \in [s,t]} |\theta(r)|$.

We prove statement (ii) by the contrapositive and show that the following two conditions are equivalent:

  **C1**: $\mathrm{HF}_0(s, t) = \min_{r \in [s,t]} |\theta(r)|$.

  **C2**: $\exists r \in [s, t]$ such that $\theta(r') \leq \theta(r), \forall r' \in [s, t]$.

Note that **C2** is equivalent to there is no 0-conflict in $[s, t]$. "$\Longleftarrow$" consider first that **C2** is true and $\theta(r)$ is an upper bound for the partitions $\theta(r')$, $r' \in [s, t]$. This implies that $\forall r' \in [s, t]$ we have that $\forall C' \in \theta(r')$ there $\exists C \in \theta(r)$ such that $C' \in C$. By construction of the MCbiF (Eq. 1) this implies $\forall \sigma' \in K^{s,t}$ there $\exists \sigma \in K^{r,r}$ such that $\sigma' \subseteq \sigma$. This means $K^{s,t} \subseteq K^{r,r}$ and thus $K^{s,t} = K^{r,r}$. As $K^{r,r}$ has $|\theta(r)|$ disconnected components this implies $\mathrm{HF}_0(s, t) = |\theta(r)|$, showing **C1**.

"$\Longrightarrow$" To prove the other direction, assume that **C1** is true. Then there exists $r \in [s, t]$ such that $c := \mathrm{HF}_0(s, t) = |\theta(r)|$ with $|\theta(r)| = \min_{q \in [s,t]} |\theta(q)|$. In particular, the disconnected components of $K^{s,t}$ are given by the clusters of $\theta(r)$ denoted by $C_1, \ldots, C_c$. Let $r' \in [s, t]$ and $C' \in \theta(r')$. Then $\exists i \in [1, \ldots, c]$ such that $C' \subseteq C_i \in \theta(r)$ because otherwise the solid simplex $\Delta C'$ would connect two solid simplices in $\{\Delta C_1, \ldots, \Delta C_c\}$, contradicting that they are disconnected in $K^{s,t}$. Hence, the clusters of $\theta(r')$ are all subsets of cluster of $\theta(r)$, implying $\theta(r') \leq \theta(r)$. As $r' \in [s, t]$ was chosen arbitrary this shows **C2**.

We finally prove statement (iii). "$\Longrightarrow$" Note that $\mathrm{HF}_0(s, t) = |\theta(r)|$ implies $|\theta(r)| = \min_{r' \in [s,t]} |\theta(r')|$ according to (i). Then (ii) shows that **C2** is true for $r$, i.e., $\theta(r)$ is the maximum of the subposet $\theta([s, t])$. "$\Longleftarrow$" The other direction follows directly from the proof of (ii).

$\square$

We next prove Corollary 9 about some properties of the average 0-conflict, which follows immediately from Proposition 7.

*Proof of Corollary 9.* We begin with the proof of statement (i). If $\theta$ is hierarchical in $[s, t]$ then $\theta$ is either coarse- or fine-graining. Assume first that $\theta$ is coarse-graining, then $\theta(s') \leq \theta(t)$ for all $s' \in [s, t]$ and together with hierarchy, this implies that $\theta(t)$ is an upper bound of the subposet $\theta([s, t])$. Hence, Proposition 7 (iii) shows that $\mathrm{HF}_0(s, t) = |\theta(t)|$. Moreover, $\mathrm{HF}_0(s, t) = \min(|\theta(s)|, |\theta(t)|)$ because coarse-graining implies $|\theta(s)| \geq |\theta(t)|$. A similar argument also shows $\mathrm{HF}_0(s, t) = |\theta(s)| = \min(|\theta(s)|, |\theta(t)|)$ if $\theta$ is fine-graining.

We continue with proving (ii). $\bar{c}_0(\theta) > 0$ is equivalent to $\exists s, t \in [t_1, t_M]$ such that $\mathrm{HF}_0(s, t) < \min_{r \in [s,t]} |\theta(r)|$, according to Definition 8. This is again equivalent to $\exists s, t \in [t_1, t_M]$ such that $\theta$ has a 0-conflict in $[s, t]$, according to Proposition 7 (ii).

We finally prove statement (iii). "$\Longrightarrow$" $\bar{c}_0(\theta)$ means that $\theta$ has no 0-conflict in $[t_1, \infty)$. As $\theta$ is also coarse- or finge-graining, Proposition 6 (iii) then shows that $\theta$ is strictly hierarchical. "$\Longleftarrow$" If $\theta$ is strictly hierarchical, then it has no 0-conflicts according to Proposition 6 (ii) and statement (ii) implies that $\bar{c}_0(\theta) = 0$.

$\square$

We can further detect triangle 0-conflicts by analysing the graph-theoretic properties of the MCbiF 1-skeleton $K_1^{s,t}$. Recall that the *clustering coefficient* C of a graph is defined as the ratio of the number of triangles to the number of paths of length 2 in the graph (Luce & Perry, 1949; Newman, 2018).

**Proposition 30.** $\mathrm{C}(K_1^{s,t}) < 1$ *iff there is a triple $x, y, z \in X$ that leads to a triangle 0-conflict for $[s, t]$, and which is not a cycle, i.e., additionally to property b) in Definition 5 we also have $\nexists r_3 \in [s, t]$: $x \sim_{r_3} z$.*

*Proof of Proposition 30.* Assume that $\mathcal{C}(K_1^{s,t}) < 1$. Then there exist $x, y, z \in X$ that form a path of length 2 but no triangle, see Newman (2018) for details on the clustering coefficient. Without loss of generality, $[x, y], [y, z] \in K_1^{s,t}$ but $[x, z] \notin K_1^{s,t}$. This implies $\exists r_1, r_2 \in [s, t]$: $x \sim_{r_1} y \sim_{r_2} z$ and $\nexists r \in [s, t]$: $x \sim_r z$. Hence, $x, y, z$ lead to a triangle 0-conflict. $\qquad\square$

Let us consider the graph generated as the disjoint union of all clusters from partitions in [s,t] as cliques. This graph is equivalent to the MCbiF 1-skeleton $K_1^{s,t}$. Proposition 30 shows that the clustering coefficient of this graph can be used to detect triangle 0-conflicts that are not cycles. To be able to detect triangle 0-conflicts that correspond to non-bounding cycles, we turn to the 1-dimensional homology.

We next prove Proposition 11 on 1-conflicts.

*Proof of Proposition 11.* Statement (i) follows directly from the definition of 1-conflicts that $\mathrm{HF}_1(s, t) = \dim[H_k(K^{s,t})] \geq 1$ iff $\theta$ has a 1-conflict.

We next prove statement (ii): If $\mathrm{HF}_1(s, t) \geq 1$ there exists a 1-cycle $z = [x_1, x_2] + \cdots + [x_{n-1}, x_n] + [x_n, x_1]$ that is non-bounding, i.e., $h := [z] \neq 0$ in $H_1(K^{s,t})$, see Appendix E.3 for details. Case 1: Assume $\nexists r \in [s, t]$: $x_1 \sim_r x_2 \sim_r x_3$, then it follows immediately that $x_1, x_2, x_3$ lead to a triangle 0-conflict. Case 2: Assume $\exists r \in [s, t]$: $x_1 \sim_r x_2 \sim_r x_3$. As $[z] \neq 0$ there exists a 1-cycle $\tilde{z} = [\tilde{x}_1, \tilde{x}_2] + \cdots + [\tilde{x}_{m-1}, \tilde{x}_m] + [\tilde{x}_m, \tilde{x}_1] \in Z_1(K^{s,t})$ such that $\tilde{z}$ is homologous to $z$, i.e., $\tilde{z} = z + \partial_2 w$ for $w \in C_2(K^{s,t})$, and such that $\nexists r \in [s, t]$: $\tilde{x}_1 \sim_r \tilde{x}_2 \sim_r \tilde{x}_3$. In particular, $\tilde{x}_1, \tilde{x}_2, \tilde{x}_3$ lead to a triangle 0-conflict.

We finally prove statement (iii): If $\theta$ is hierarchical, then it has no 0-conflicts according to Corollary 9. Hence, $\theta$ also has no triangle 0-conflict in $[s, t]$ and so (i) implies that $\mathrm{HF}_1(s, t) = 0$. $\qquad\square$

**Remark 31.** *Proposition 11 states that every 1-conflict is a triangle 0-conflict. However, not every (triangle) 0-conflict is a 1-conflict, see Example 37. Note also that several triangle 0-conflicts in the sequence $\theta$ can lead to a 1-conflict, when the triangle 0-conflicts are linked together in such a way as to form a non-bounding cycle, see Example 37. We can test for these systematically using $\mathrm{HF}_1$.*

**Remark 32.** *While 0-conflicts ($\bar{c}_0(\theta) > 0$) can be defined in relation to the refinement order that gives rise to the partition lattice, the partition lattice cannot be used to detect higher-order cluster inconsistencies (1-conflicts), which can be captured and quantified instead by $\mathrm{HF}_1$ and the average measure $\bar{c}_1(\theta)$.*

### B.2.2 PROOFS AND DETAILS FOR SECTION 3.2

To prove the equivalence between the MPH of the MCbiF and nerve-based MPH, we can use a Persistent Nerve Lemma for abstract simplicial complexes by Schindler & Barahona (2025).

**Lemma 33** (Schindler & Barahona (2025), Lemma 32). *Let $K \subseteq K'$ be two finite abstract simplicial complexes and $\{K_\alpha\}_{\alpha \in A}$ and $\{K'_\alpha\}_{\alpha \in A}$ be subcomplexes that cover $K$ and $K'$ respectively, based on the same finite parameter set such that $K_\alpha \subseteq K'_\alpha$ for all $\alpha \in A$. Let $\tilde{K}$ denote the nerve $\mathcal{N}(\{|K_\alpha|\}_{\alpha \in A})$ and $\tilde{K}'$ the nerve $\mathcal{N}(\{|K'_\alpha|\}_{\alpha \in A})$. If the intersections $\bigcap_{i=0}^k |K_{\alpha_i}|$ and $\bigcap_{i=0}^k |K'_{\alpha_i}|$ are either empty or contractible for all $k \in \mathbb{N}$ and for all $\alpha_0, ..., \alpha_k \in A$, then there exist homotopy equivalences $\tilde{K} \to |K|$ and $\tilde{K}' \to |K'|$ that commute with the canonical inclusions $|K| \hookrightarrow |K'|$ and $\tilde{K} \hookrightarrow \tilde{K}'$.*

See Schindler & Barahona (2025, Lemma 32) for a proof.

Next, we provide the proof about the equivalence between MCbiF and nerve-based MCbiF.

**Proposition 34.** *The bifiltrations $\mathcal{M}$ and $\tilde{\mathcal{M}}$ lead to the same persistence module.*

*Proof.* We use Lemma 33 to show the equivalence between MCbiF and nerve-based MCbiF. For $s \geq s'$ and $t \leq t'$, we denote $\tilde{K} := \tilde{K}^{s,t}$ and $K := K^{s,t}$, and further denote $\tilde{K}' := \tilde{K}^{s',t'}$ and $K' := K^{s',t'}$. As $\theta$ is a piecewise-constant function with change points $t_1 < \cdots < t_M$, we can assume without loss of generality that $s' = t_k$, $s = t_\ell$, $t = t_m$, $t' = t_n$ for change points $t_k < t_\ell < t_m < t_n$ of $\theta$. For the multi-index set $A := A(k, n)$, define the cover $\{K_\alpha\}_{\alpha \in A}$ by $K_\alpha = \Delta C_\alpha$ if $\alpha \in A(\ell, m) \subseteq A$ and $K_\alpha = \emptyset$ otherwise and the cover $\{K'_\alpha\}_{\alpha \in A}$ by $K'_\alpha = \Delta C_\alpha$

for all $\alpha \in A$. Then we have $K_\alpha \subseteq K'_\alpha$ for all $\alpha \in A$ and we recover the MCbiF $K = \bigcup_{\alpha \in A} K_\alpha$ and the nerve-based MCbiF $\tilde{K} = \mathcal{N}(\{K'_\alpha\}_{\alpha \in A})$ and similarly we recover $K'$ and $\tilde{K}'$. For any $k \in \mathbb{N}$ and $\alpha_0, ..., \alpha_k \in A$, the intersections $\bigcap_{i=0}^k |K'_{\alpha_i}|$ are intersections of solid simplices and thus either empty or contractible. For a detailed proof of this fact, see the proof of Proposition 30 in Schindler & Barahona (2025). Lemma 33 now yields homotopy equivalences $\tilde{K} \to |K|$ and $\tilde{K}' \to |K'|$ that commute with the canonical inclusions $|K| \hookrightarrow |K'|$ and $\tilde{K} \hookrightarrow \tilde{K}'$. This shows the equivalence between the MPH of the MCbiF and the MPH of the nerve-based MCbiF. $\square$

Next, we prove Proposition 16 about the dimension of the nerve-based MCbiF.

*Proof of Proposition 16.* Statement (i) follows directly from the definition in Eq. equation 1. We show statement (ii) by induction. Base case: From the definition of the nerve-based MCbiF, it follows directly that $\dim N^{t_m, t_m} = 0$ because the indices in $A(m, m)$ correspond to mutually exclusive clusters. Induction step: Let us assume that $\dim N^{t_m, t_{m+n}} = n$, then there exist $C_0, \ldots, C_n \in \theta([t_m, t_{m+n}])$ such that $C_0 \cap \cdots \cap C_n \neq \emptyset$. As the clusters in partition $\theta(t_{m+n+1})$ cover the set $X$ there exist a cluster $C \in \theta(t_{m+n+1})$ such that $C \cap C_0 \cap \cdots \cap C_n \neq \emptyset$. Hence, $\dim N^{t_m, t_m+n} \geq n + 1$. If $\dim N^{t_m, t_m+n} > n + 1$ there would exist a second cluster $C' \in \theta(t_{m+n+1})$ with $C' \cap C \cap C_0 \cap \cdots \cap C_n \neq \emptyset$ but $C' \cap C \neq \emptyset$ contradicts that clusters of $\theta(t_{m+n+1})$ are mutually exclusive. Hence, $\dim N^{t_m, t_m+n} = n + 1$, proving statement (ii) by induction. $\square$

We provide a proof for the connection between Sankey diagrams and the nerve-based MCbiF.

*Proof of Proposition 17.* The Sankey diagram graph $S(\theta) = (V = V_1 \uplus ... \uplus V_M, E = E_1 \uplus ... \uplus E_{M-1})$ is a strict 1-dimensional subcomplex of $\tilde{K} = \tilde{K}^{t_1, t_M}$ because $\tilde{K}^{t_m, t_m} = V_m \subseteq \tilde{K}$ and $\tilde{K}^{t_m, t_{m+1}} = E_m \subseteq \tilde{K}$. This also shows that the zigzag filtration equation 5 contains exactly the same vertices (0-simplices) and edges (1-simplices) as $S(\theta)$. $\square$

We next prove Proposition 18 that characterises conflicts that can arise in a single layer of the Sankey diagram.

*Proof of Proposition 18.* (i) Suppose that $\theta$ has a 0-conflict in $[t_m, t_{m+1}]$. Then $\theta(t_m) \not\leq \theta(t_{m+1})$ and $\theta(t_m) \not\geq \theta(t_{m+1})$. This means that there exists $C \in \theta(r_1)$ such that $\exists C', C'' \in \theta(r_2)$ with $C \cap C' \neq \emptyset$, $C \cap C'' \neq \emptyset$ and $C' \cap C'' = \emptyset$, otherwise $\theta(r_1) \leq \theta(r_2)$. Hence, $\theta(r_1) \not\leq \theta(r_2)$ is equivalent to $\exists u \in V_m$ (the node corresponding to cluster $C$) with degree $\deg(u) \geq 2$ in $E_m$. An analogous argument shows that $\theta(r_1) \not\geq \theta(r_2)$ is equivalent to $\exists v \in V_{m+1}$ with $\deg(v) \geq 2$. This proves the statement.

(ii) Let $x, y, z \in X$ form a triangle 0-conflict for the interval $[t_m, t_{m+1}]$, i.e., $x \sim_{t_m} y \sim_{t_{m+1}} z$ but $x \not\sim_{t_{m+1}} y \not\sim_{t_m} z$. In particular, the elements $x, y, z$ are mutually distinct. This means there exist $C_1, C_2 \in \theta(t_m)$ and $C'_1, C'_2 \in \theta(t_{m+1})$ such that $x, y \in C_1$, $z \in C_2$, $y, z \in C'_1$ and $x \in C'_2$. This is equivalent to $C_1 \cap C_2 = \emptyset$, $C'_1 \cap C'_2 = \emptyset$ and $C_1 \cap C'_1 \neq \emptyset$, $C_1 \cap C'_2 \neq \emptyset$, $C_2 \cap C'_2 \neq \emptyset$. Let $u, u' \in V_m$ correspond to $C_1$ and $C_2$, respectively, and $v, v' \in V_{m+1}$ correspond to $C'_1$ and $C'_2$, respectively. Then the above is equivalent to $[u', v], [v, u], [u, v'] \in E_m$, which is again equivalent to the existence of a path in $E_m$ that has length at least 3.

(iii) The statement follows from the fact that every cycle in $E_m$ is even because the graph $(V_m \uplus V_{m+1}, E_m)$ is bipartite and the fact that every cycle in $E_m = K^{t_m, t_{m+1}}$ is non-bounding because $\dim K^{t_m, t_{m+1}} = 1$. $\square$

### B.3 PROOFS AND DETAILS FOR SECTION 4

We continue by proving that 0-conflicts can induce violations of the strong triangle inequality as stated in Proposition 21.

*Proof of Proposition 21.* Let $x, y, z \in X$ lead to a triangle 0-conflict for the interval $[s, t]$, i.e., $\exists r_1, r_2 \in [s, t]$: $x \sim_{r_1} y \sim_{r_2} z$ and $\nexists r \in [s, t]$: $x \not\sim_r y \not\sim_r z$. This means $D_{\theta, s}(x, y) \leq r_1$

and $D_{\theta,s}(y,z) \leq r_2$. Let us define $r_3 := D_{\theta,s}(x_z) \leq t_M$. We know that $r_3 \neq r_1$ and $r_3 \neq r_2$ as otherwise $x \sim_{r_3} y \sim_{r_3} z$ for $r_3 \in [s,t]$, contradicting the lack of transitivity. Hence, without loss of generality, $r_1 < r_2 < r_3$. Then the above is equivalent to $D_{\theta,s}(x_z) = r_3 > r_1 \geq \min(D_{\theta,s}(x,y), D_{\theta,s}(y,z))$, which is a violation of the strong triangle inequality. $\square$

Next, we state Proposition 35 that establishes the connection between the MPH of the MCbiF and that of the Merge-Rips bifiltration constructed from the matrix of first-merge times, $D_{\theta,s}$.

**Proposition 35.** *Let us define the Merge-Rips bifiltration $\mathcal{L}$ based on $D_{\theta,s}$ as*

$$\mathcal{L} = (L^{s,t})_{t_1 \leq s \leq t} \quad where \quad L^{s,t} = \{\sigma \subset X \mid \forall x,y \in \sigma : D_{\theta,s}(x,y) \leq t\}. \tag{8}$$

*Then the 0-dimensional MPH of the Merge-Rips bifiltration, $\mathcal{L}$, and of the MCbiF, $\mathcal{M}$, are equivalent, but the 1-dimensional MPH of $\mathcal{L}$ and $\mathcal{M}$ are generally not equivalent. Furthermore, if $\theta$ is strictly hierarchical, then $\mathcal{L}$ has a trivial 1-dimensional MPH.*

*Proof of Proposition 35.* First note that $\mathcal{L} = (L^{s,t})_{t_1 \leq s \leq t}$ is indeed a well-defined bifiltration because $L^{s,t} \subseteq L^{s',t'}$ if $s \geq s'$ and $t \leq t'$. In particular, $\mathcal{L}$ is also defined uniquely on the finite grid $P = \{(s,t) \in [t_1, \ldots, t_M] \times [t_1, \ldots, t_M] \mid s \leq t\}$ with partial order $(s,t) \leq (s',t')$ if $s \geq s', t \leq t'$.

The proof of the proposition then follows from a simple extension of Proposition 36 in Schindler & Barahona (2025) to the 2-parameter case. To see that the 0-dimensional MPH of $\mathcal{L}$ and $\mathcal{M}$ are equivalent, note that both bifiltrations have the same 1-skeleton. Moreover, the 1-dimensional MPH is generally not equivalent because $\mathcal{L}$ is a Rips-based bifiltration and thus 2-determined, whereas $\mathcal{M}$ is not 2-determined.

If $\theta$ is strictly hierarchical, then $D_{\theta,s}$ fulfils the strong-triangle inequality, and thus the Rips-based bifiltration leads to a trivial 1-dimensional homology, see (Schindler & Barahona, 2025, Corollary 37). Hence, the 1-dimensional MPH of $\mathcal{L}$ is trivial. $\square$

Finally, we provide a brief proof for Proposition 22 linking the 0-dimensional Hilbert function of a pair of partitions and the graph Laplacian built from the conditional entropy matrix between both partitions.

*Proof of Proposition 22.* Using Proposition 17, we prove the statement with the equivalent nerve-based MCbiF. Note that the graph $G := P_{t_2|t_1} P_{t_2|t_1}^T$ has the same vertices and edges as the simplicial complex $\tilde{K}^{t_1,t_2}$, which is 1-dimensional and thus also a graph according to Proposition 16. This shows that $\mathrm{HF}_{(}t_1, t_2)$ is given by the number of connected components in $G$. Furthermore, observe that $P_{t_2|t_1}^T \mathbf{1} = \mathbf{1}$, and that the resulting matrix $L$ is the Laplacian of the undirected graph $G$. Hence, $\dim(\ker L)$ is equal to the number of connected graph components (Chung, 1997), proving the statement. $\square$

### B.4 PROOFS AND DETAILS FOR SECTION 6

It follows from the construction of MCbiF that the Hilbert functions are invariant to certain swaps of partitions in $\theta$.

**Proposition 36.** $\mathrm{HF}_k(s,t)$ *is invariant to swaps of partitions in sequence $\theta$ between $s$ and $t$, for $t_1 \leq s \leq t$.*

*Proof.* Let us denote the change points of $\theta$ by $t_1 < t_2 < \cdots < t_M$. Without loss of generality, $s = t_m$ and $t = t_{m+n}$ for $m + n \leq M$. Let us now consider a permutation $\tau : [1, \ldots, M] \rightarrow [1, \ldots, M]$ such that $\tau(i) = i$ for $1 \leq i < m$ and $m + n < i \leq M$ and define the permuted sequence of partitions $\theta_\tau$ as $\theta_\tau(t_m) = \theta(t_{\tau(m)})$. Despite the permutation we still get the same MCbiF for $\theta$ and $\theta_\tau$ for parameters $s \leq t$ because

$$\bigcup_{s \leq r \leq t} \bigcup_{C \in \theta(r)} \Delta C = \bigcup_{s \leq r \leq t} \bigcup_{C \in \theta_\tau(r)} \Delta C.$$

This implies that $\mathrm{HF}_k(s,t)$ is the same for $\theta$ and $\theta_\tau$. $\square$

## C ADDITIONAL EXAMPLES

Our first example corresponds to the sequence of partitions analysed in Fig.1.

**Example 37** (3-element example). *Let $X = \{x_1, x_2, x_3\}$ and we define $\theta(0) = \hat{0}$, $\theta(1) = \{\{x_1, x_2\}, \{x_3\}\}$, $\theta(2) = \{\{x_1\}, \{x_2, x_3\}\}$, $\theta(3) = \{\{x_1, x_3\}, \{x_2\}\}$ and $\theta(4) = \hat{1}$ so that $\theta$ is coarse-graining with $M = 5$ change points. This example corresponds to Fig. 1a, and the 0- and 1-dimensional Hilbert functions are provided in Fig. 1b.*

*Note that $\mathrm{HF}_0(1, 2) < |\theta(2)|$ and $\mathrm{HF}_0(2, 3) < |\theta(3)|$ indicates the presence of two triangle 0-conflicts, which are not 1-conflicts because $\mathrm{HF}_1(1, 2) = \mathrm{HF}_1(2, 3) = 0$. See Fig. 7a for an illustration. As shown in Proposition 21, the triangle 0-conflicts violate the strong triangle inequality of the matrix of first merge times $D_\theta$ (Eq 6), e.g., $D_\theta(x_1, x_3) = 3 > \max(D_\theta(x_1, x_2), D_\theta(x_2, x_3)) = 2$.*

*In addition, $\mathrm{HF}_1(1, 3) = 1$ indicates the presence of a 1-conflict that arises from the higher-order inconsistencies of cluster assignments across partitions $\theta(1)$, $\theta(2)$ and $\theta(3)$. See Fig. 7b for an illustration. In particular, the equivalence relations $x_1 \sim_1 x_2$, $x_2 \sim_2 x_3$ and $x_3 \sim_3 x_1$ induce a 1-cycle $z = [x_1, x_2] + [x_2, x_3] + [x_3, x_1] \in Z_1(K^{1,3})$ and due to the lack of transitivity on the interval $[1, 3]$, the 1-cycle $z$ is also non-bounding, yielding a 1-conflict. The 1-conflict then gets resolved at $t = 4$ because $\theta(4) = \hat{1}$ restores transitivity on the interval $[1, 4]$. See Fig. 7c for an illustration.*

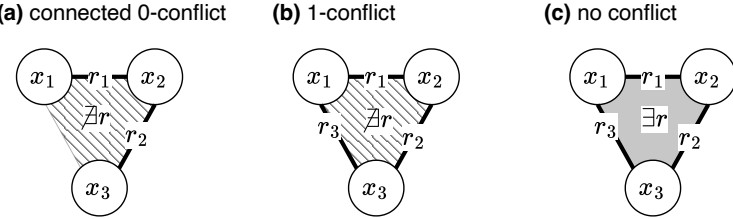

Figure 7: (a) Illustration of a triangle 0-conflict that violates the strong triangle inequality of the matrix of first merge times $D_{\theta,s}$ (Eq 6), (b) a 1-conflict and (c) three elements that are in no conflict due to global transitivity. If we choose $r_1 = 1, r_2 = 2, r_3 = 3$ and $r = 4$, the conflicts depicted here correspond to the conflicts in Example 37.

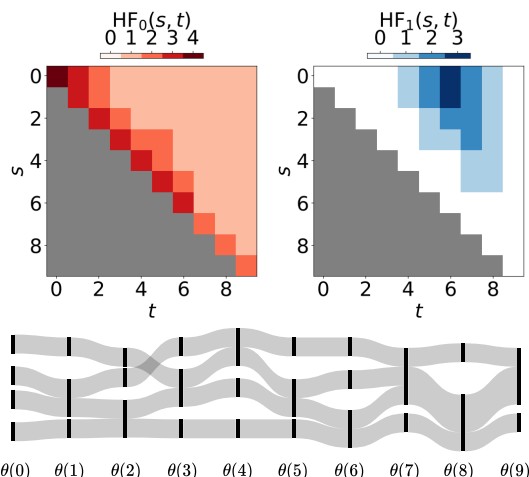

Figure 8: Hilbert functions and optimised Sankey diagram for the sequence of partitions $\theta$ defined in Example 38.

**Example 38** (4-element example). *We now consider the more complex case of a 4-element set $X = \{x_1, x_2, x_3, x_4\}$. Let us start with $\theta(0) = \hat{0}$ and append in sequence the 6 distinct partitions that contain two singletons and one cluster of size 2, i.e., $\theta(1) = \{\{x_1, x_2\}, \{x_3\}, \{x_4\}\}$, $\theta(2) = \{\{x_1\}, \{x_2, x_3\}, \{x_4\}\}$, $\theta(3) = \{\{x_1\}, \{x_2\}, \{x_3, x_4\}\}$, $\theta(4) = \{\{x_1, x_3\}, \{x_2\}, \{x_4\}\}$, $\theta(5) = \{\{x_1, x_4\}, \{x_2\}, \{x_3\}\}$ and $\theta(6) = \{\{x_1\}, \{x_2, x_4\}, \{x_3\}\}$. Finally, we append consecutively three*

*partitions, each of which contains a cluster of size 3, i.e., $\theta(7) = \{\{x_1, x_2, x_3\}, \{x_4\}\}$, $\theta(8) = \{\{x_1\}, \{x_2, x_3, x_4\}\}$ and $\theta(9) = \{\{x_1, x_3, x_4\}, \{x_2\}\}$. $\theta$ is a coarse-graining, non-hierarchical sequence with $M = 10$ change points. See Fig. 8 for a Sankey diagram of $\theta$.*

*To analyse the topological autocorrelation of $\theta$, we compute the MCbiF Hilbert functions $\mathrm{HF}_k$ for dimensions $k = 0, 1$ (see Fig. 8). We observe that $\mathrm{HF}_0(s, s + 3) = 1 < \min_{r \in [s, s+3]} |\theta(r)|$ for all $0 \le s \le 8$, which implies that the hierarchy of $\theta$ is broken after no-less than three steps in the sequence when starting at scale $s$. Moreover, we can detect that $\theta$ is non-nested and has higher-order cluster inconsistencies because 1-conflicts emerge at scales $t = 4, 5, 6$, as indicated by non-zero values in $\mathrm{HF}_1$. The 1-conflicts get resolved one-by-one through the partitions that contain clusters of size 3, and at $t = 9$, when the third such partition appears in $\theta$, all 1-conflicts are resolved.*

Finally, we show an example that demonstrates how conditional entropy does not detect 1-conflicts in general.

**Example 39** (CE cannot detect 1-conflicts). *Let $X = \{x_1, x_2, x_3, x_4\}$ for which we consider two different sequences of partitions $\theta(t)$ and $\eta(t)$ such that $\theta(1) = \eta(1) = \{\{x_1, x_2\}, \{x_3\}, \{x_4\}\}$, $\theta(2) = \eta(2) = \{\{x_1\}, \{x_2, x_3\}, \{x_4\}\}$ but $\theta(3) = \{\{x_1, x_3\}, \{x_2\}, \{x_4\}\} \ne \theta(3) = \{\{x_1\}, \{x_2\}, \{x_3, x_4\}\}$. See Fig. 9 for a Sankey diagram representation of the two sequences of partitions. Note that $\theta$ and $\eta$ only differ at scale $t = 3$. However, this difference is crucial because a 1-conflict emerges in $\theta$ at scale $t = 3$, whereas $\eta$ has only triangle 0-conflicts and no 1-conflict. Note that $\theta$ corresponds to the toy example in Fig. 1 with one additional isolated element.*

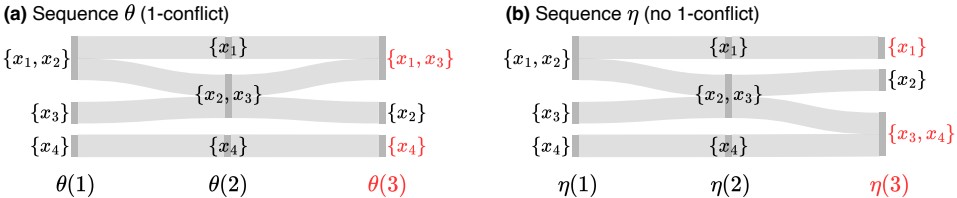

Figure 9: Sankey diagrams for sequences $\theta$ and $\eta$ defined in Example 39. Note that a 1-conflict emerges in $\theta$ at scale $t = 3$, but $\eta$ has no 1-conflict.

*In accordance with our theoretical results developed in Section 3.1, we can use the 1-dimensional Hilbert function $\mathrm{HF}_1$ to detect the 1-conflict in $\theta$ and distinguish the two sequences. In particular, $\mathrm{HF}_1(\theta(1), \theta(3)) = 1$ but $\mathrm{HF}_1(\eta(i), \eta(j)) = 0$ for all $i, j \in [1, 2, 3]$, $i \le j$. In contrast, the conditional entropy $\mathrm{H}$ (see Eq. 16) cannot distinguish between the two sequences as they yield the same pairwise conditional entropies. In particular, $\mathrm{H}(\theta(i)|\theta(j)) = \mathrm{H}(\eta(i)|\eta(j)) = \frac{1}{2} \log 2$ for $i \ne j$. This demonstrates that the conditional entropy cannot detect higher-order cluster inconsistencies in sequences of partitions.*

# D DETAILS ON EXPERIMENTS

## D.1 REGRESSION TASK

We first provide a rigorous definition of the space of coarse-graining sequences of partitions.

**Definition 40** (Space of coarse-graining sequences of partitions). *The space of coarse-graining sequences of partitions, denoted $\Pi_N^M$, is defined as the set of coarse-graining sequences $\theta : [0, \infty) \to \Pi_X$ with $|X| = N$ and $M$ change points $t_m = 0, \ldots, M - 1$, such that $|\theta(s)| \ge |\theta(t)|$, $\forall s \le t$, which start with the finest partition $\theta(t_1 = 0) = \hat{0}$ and end with the full set $\theta(t_M = M - 1) = \hat{1}$.*

For our experiments we sample randomly from the spaces $\Pi_5^{20}$ and $\Pi_{10}^{20}$. Note that $|\Pi_N^1|$ is given by the exponentially growing Bell numbers $B_N$ with $B_{10} = 115, 975 \gg B_5 = 52$ Stanley (2011).

Figure 10 shows the correlation between the minimal crossing number $y = \overline{\kappa}(\theta)$ (Eq. 11) and summary statistics of the five feature maps under investigation: the consensus indices $\overline{\mathrm{VI}}$, $\overline{\mathrm{ARI}}$ and $\overline{\mathrm{MOD}}$, and the MCbiF topological average measures $\bar{c}_0$ and $\bar{c}_1$. In addition to the results already described in the main text, we also observe that the correlation between $\overline{\mathrm{VI}}$ and $\bar{c}_0$ ($r = -0.32$ for

$N = 5$, $r = -0.48$ for $N = 10$) is stronger than with $\bar{c}_1$ ($r = -0.12$ for $N = 5$, $r = -0.34$ for $N = 10$). This can be explained by the fact that $\overline{\text{VI}}$ and $\bar{c}_0$ can both be computed from pairwise interactions of clusters in contrast to $\bar{c}_1$, see Section 4. Furthermore, we observe a strong correlation between $\bar{c}_0$ and $\bar{c}_1$ ($r = 0.52$ for $N = 5$ and $r = 0.43$ for $N = 10$) because of the dependencies between 0- and 1-conflicts, see Section 3.1

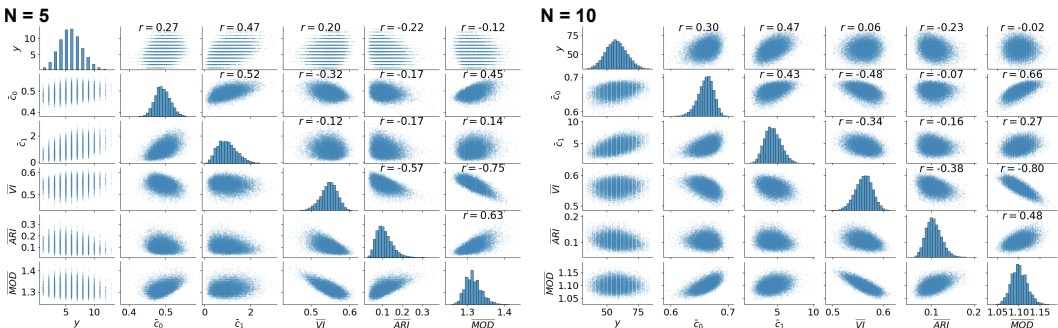

Figure 10: Pearson correlation ($r$) between crossing number $y$, consensus indices $\overline{\text{VI}}$, $\overline{\text{ARI}}$ and $\overline{\text{MOD}}$, and MCbiF-based conflict measures $\bar{c}_0$ and $\bar{c}_1$ for $N = 5$ and $N = 10$.

Note that we can consider our five feature maps as $M \times M$ greyscale images, where $\text{HF}_0$, $\text{HF}_1$, ARI and MOD are symmetric and CE is asymmetric. The raw label encoding of $\theta$ is also similarly interpreted as an $N \times M$ greyscale image. For our regression task, we train a simple CNN (LeCun & Bengio, 1998) with one convolution and max-pool layer and one fully connected layer and also a simple MLP (Bishop, 2006) on the flattened images with one or two hidden layers and dropout (Srivastava et al., 2014). For each feature map (or their combinations) separately, we perform hyperparameter optimisation for the number of filters (ranging from 2 to 6) and kernel size (chosen as 4, 8, 16, 32 or 64) in the CNN and the number of nodes (chosen as 4, 8, 16, 32, 64, 128 or 256), number of layers (1 or 2) and dropout rate (chosen as 0.00, 0.25 or 0.50) in the MLP. We use the Adam optimiser (Kingma & Ba, 2017) with batch size 32 and learning rate chosen as 0.01, 0.005, 0.001, 0.0005 or 0.0001 for training. We train both MLP and CNN over 150 epochs with early-stopping and a patience of 10 epochs.

As an additional baseline model, we train a graph convolutional network (GCN) (Kipf & Welling, 2017) with a regression head on the weighted adjacency matrices of the Sankey diagram graph $S(\theta)$ (Eq. 9), where the edges are weighted according to the number of elements in the overlap of two clusters. We choose three-dimensional node features consisting of a constant, the normalised weighted degree and the layer number of the node in the Sankey diagram. We perform hyperparameter optimisation for the number of hidden dimensions (16, 32, 64, 128), number of layers (1, 2 or 3) and dropout rate (0, 0.2 or 0.5). We use again Adam optimiser with batch size 32, learning rate chosen as 0.01, 0.005, 0.001, 0.0005 or 0.0001 and weight decay chosen as 0.0 or 0.0001, as well as the `ReduceLROnPlateau` learning rate scheduler,[2] and train the GCN for 150 epochs with early-stopping and a patience of 10 epochs, consistent with the other models.

We perform a full grid search of the hyperparameter space for MLP and CNN trained on the different feature maps (or their combinations). We perform 145 trials of hyperparameter search with the Tree-Structured Parzen Estimator (TPE) (Watanabe, 2025) for the GCN, as a full grid search was prohibited by increased computational complexity. We used the train split of our data for training and the validation split for evaluation and hyperparameter selection. Below, we detail the hyperparameters for the best models trained on the different features, which were chosen according to the performance on the validation split. We first report details for $N = 5$:

- Optimal model for raw label encoding at $N = 5$: CNN with 16 filters, kernel size 4 and learning rate 0.001.
- Optimal model for Sankey graph $S(\theta)$ at $N = 5$: GCN with three hidden layers of dimension 128 each, no dropout, no weight decay and learning rate 0.01.

---

[2]`https://docs.pytorch.org/docs/stable/generated/torch.optim.lr_scheduler.ReduceLROnPlateau.html`

Table 3: Regression task. Test $R^2$ score with bootstrapped 95% confidence intervals of LR, CNN, MLP and GCN models trained on different feature sets for $N = 5$ and $N = 10$.

| N | Method | Raw label encoding | Sankey graph | $HF_0$ | $HF_1$ | $HF_0$ & $HF_1$ | CE | ARI | MOD |
|---|---|---|---|---|---|---|---|---|---|
| 5 | LR | 0.078 (0.060–0.095) | - | 0.147 (0.125–0.168) | 0.486 (0.463–0.509) | 0.539 (0.518–0.559) | 0.392 (0.367–0.414) | 0.166 (0.142–0.188) | 0.413 (0.388–0.436) |
| | CNN | 0.267 (0.241–0.293) | - | 0.155 (0.133–0.176) | 0.504 (0.480–0.526) | **0.544** (0.522–0.565) | 0.492 (0.468–0.514) | 0.422 (0.397–0.446) | 0.354 (0.327–0.280) |
| | MLP | 0.104 (0.083–0.124) | - | 0.150 (0.130–0.170) | 0.491 (0.469–0.513) | 0.541 (0.520–0.561) | 0.409 (0.385–0.432) | 0.214 (0.191–0.236) | 0.351 (0.325–0.375) |
| | GCN | - | 0.416 (0.392–0.438) | - | - | - | - | - | - |
| 10 | LR | 0.038 (0.022–0.053) | - | 0.214 (0.191–0.237) | 0.448 (0.425–0.470) | **0.516** (0.494–0.537) | 0.457 (0.434–0.479) | 0.246 (0.221–0.270) | 0.345 (0.320–0.369) |
| | CNN | 0.072 (0.050–0.092) | - | 0.211 (0.188–0.233) | 0.448 (0.425–0.470) | 0.507 (0.486–0.527) | 0.454 (0.431–0.476) | 0.294 (0.268–0.319) | 0.312 (0.286–0.335) |
| | MLP | 0.036 (0.019–0.053) | - | 0.212 (0.191–0.232) | 0.450 (0.426–0.473) | 0.514 (0.493–0.534) | 0.458 (0.435–0.480) | 0.256 (0.230–0.280) | 0.246 (0.221–0.269) |
| | GCN | - | 0.229 (0.208–0.248) | - | - | - | - | - | - |

Table 4: Train $R^2$ scores of LR, CNN, MLP and GCN models trained on different features for $N = 5$ and $N = 10$.

| $N$ | Method | Raw label encoding | Sankey graph | $HF_0$ | $HF_1$ | $HF_0$ & $HF_1$ | CE | ARI | MOD |
|---|---|---|---|---|---|---|---|---|---|
| 5 | LR | 0.096 | - | 0.163 | 0.493 | 0.550 | 0.409 | 0.194 | 0.419 |
| | CNN | 0.321 | - | 0.170 | 0.509 | **0.562** | 0.515 | 0.464 | 0.460 |
| | MLP | 0.254 | - | 0.160 | 0.499 | 0.547 | 0.439 | 0.366 | 0.396 |
| | GCN | - | 0.397 | - | - | - | - | - | - |
| 10 | LR | 0.061 | - | 0.230 | 0.456 | **0.522** | 0.464 | 0.255 | 0.370 |
| | CNN | 0.112 | - | 0.220 | 0.456 | 0.519 | 0.476 | 0.376 | 0.368 |
| | MLP | 0.114 | - | 0.218 | 0.453 | 0.515 | 0.468 | 0.368 | 0.282 |
| | GCN | - | 0.234 | - | - | - | - | - | - |

- Optimal model for $HF_0$ & $HF_1$ at $N = 5$: CNN with 4 filters, kernel size 3, and learning rate 0.001.

- Optimal model for CE at $N = 5$: CNN with 8 filters, kernel size 2, and learning rate 0.005.

- Optimal model for ARI at $N = 5$: CNN with 8 filters, kernel size 2, and learning rate 0.005.

- Optimal model for MOD at $N = 5$: LR.

We next report the details for $N = 10$:

- Optimal model for raw label encoding at $N = 10$: CNN with 8 filters, kernel size 3 and learning rate 0.01.

- Optimal model for Sankey graph $S(\theta)$ at $N = 10$: GCN with three hidden layers of dimension 128 each, dropout rate 0.25, no weight decay and learning rate 0.005.

- Optimal model for $HF_0$ & $HF_1$ at $N = 10$: LR.

- Optimal model for CE at $N = 10$: MLP with a single layer of 256 nodes, no dropout and a learning rate of 0.001.

- Optimal model for ARI at $N = 10$: CNN with 64 filters, kernel size 2, and learning rate 0.005.

- Optimal model for MOD at $N = 10$: LR.

We present the train $R^2$ scores for the optimised LR, CNN, MLP and GCN models trained on the different features in Table 4. The test $R^2$ scores are presented in Table 1 in the main text. We also report 95% confidence intervals for the test $R^2$ score in Table 3, which were computed using bootstrapping with 5,000 iterations on the test data.

### D.2 CLASSIFICATION TASK

We first provide the definition for order-preserving sequences of partitions.

**Definition 41** (Order-preserving sequence of partitions). *When a partition $\theta(t_m)$ is equipped with a total order $<_m$ on the clusters it is called an ordered partition.[3] Such a partition induces a total preorder $\lesssim_m$ on X (Stanley, 2011), i.e., if $[x]_t <_m [y]_t$ then $x \lesssim_m y$. We call the $\theta$ order-preserving if there exist total orders $(<_1, \ldots, <_M)$ such that the total preorders $(\lesssim_1, \ldots, \lesssim_M)$ are compatible across the sequence, i.e., $\forall \ell, m$ we have $x \lesssim_\ell y$ iff $x \lesssim_m y$, $\forall x, y \in X$.*

According to this definition, a sequence $\theta$ is *non-order-preserving* if there is no total order on $X$ that is consistent with all the total preorders induced by the partitions $\theta(t)$.

---

[3]The ranking $\tau_m : V_m \to \{1, \ldots, |V_m|\}$ of the vertices $V_m$ in the Sankey diagram $S(\theta)$ is one example of a total order $<_m$ on the clusters, see Section E.1.

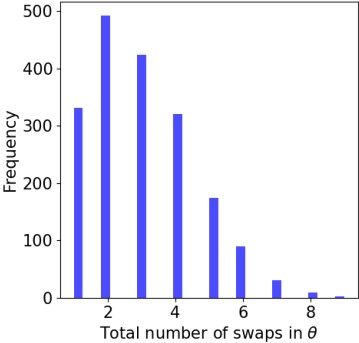

Figure 11: Classification task: Histogram of the number of swaps in non-order-preserving sequences $\theta$ (class $y = 1$). See text for the scheme to introduce random swaps in the cluster assignments as a means to break order-preservation.

**Details on Synthetic Data.** We generate order-preserving ($y = 0$) sequences $\theta \in \Pi_N^M$ through the following scheme: Let us assume that we have a total order $X = \{x_1, \ldots, x_N\}$ given by the element labels, i.e., $x_i < x_j$ if $i < j$. We construct each $\theta(t_m)$, $m = 0, \ldots, M - 1$, by cutting $X$ into clusters of the form $C = \{x_i, x_{i+1}, \ldots, x_{i+n}\}$. It is easy to verify that $\theta$ is indeed order-preserving. We adapt this scheme to generate sequences $\theta \in \Pi_N^M$ that are non-order-preserving ($y = 0$): Again, we start by constructing each sequence $\theta(t_m)$ through cutting the ordered set $X$ as before. Additionally, with probability $p = 0.1$, we swap the cluster assignments in $\theta(t_m)$ for two arbitrary elements $x, y \in X$. If $N$ and $M$ are large enough, the so-generated sequence $\theta$ is almost surely non-order-preserving. We chose $N = 500$ and $M = 30$ to demonstrate the scalability of the MCbiF method.

The number of clusters of all our generated sequences of partitions $\theta \in \Pi_N^M$ for both classes is decreasing linearly. Moreover, the average number of swaps for sequences with $y = 1$ is 2.98 for our choice of $p = 0.1$, see Fig. 11.

**Results.** We find no significant difference between the baseline consensus indices $\overline{\text{VI}}$, $\overline{\text{ARI}}$ and $\overline{\text{MOD}}$ of order-preserving ($y = 0$) and non-order-preserving ($y = 1$) sequences. In contrast, we observe a statistically significant increase of $\bar{c}_0$ and $\bar{c}_1$ for order-preserving sequences (Fig. 12).

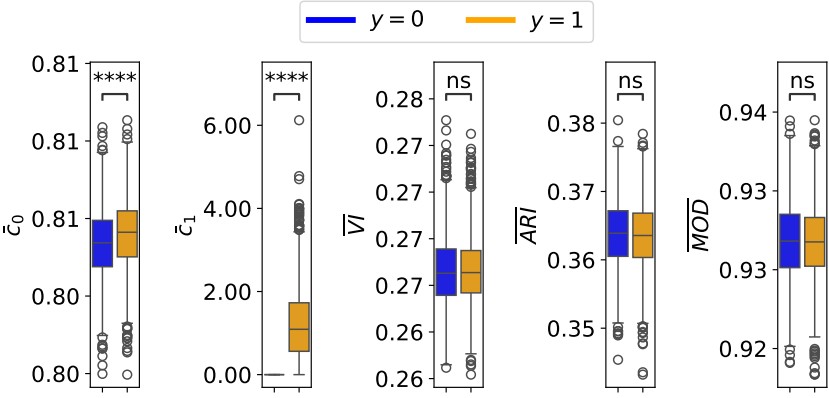

Figure 12: Difference between the baseline consensus indices $\overline{\text{VI}}$, $\overline{\text{ARI}}$ and $\overline{\text{MOD}}$ and the MCbiF topological average measures $\bar{c}_0$ and $\bar{c}_1$ of order-preserving ($y = 0$) and non-order-preserving ($y = 1$) sequences of partitions (**** indicates $p < 0.0001$, Mann-Whitney U test).

We measured the performance variability in the classification task with bootstrapped 95% confidence intervals, see Table 5, which were computed using bootstrapping with 5,000 iterations on the test data.

Table 5: Classification task. Test accuracy with bootstrapped 95% confidence intervals of logistic regression trained on different features.

| Raw label encoding | $HF_0$ | $HF_1$ | CE | ARI | MOD |
|---|---|---|---|---|---|
| 0.53 (0.50–0.57) | 0.56 (0.53–0.60) | **0.97** (0.96–0.98) | 0.50 (0.47–0.54) | 0.49 (0.45–0.53) | 0.46 (0.43–0.50) |

### D.3 APPLICATION TO REAL-WORLD TEMPORAL DATA

**Data Preprocessing.** The temporal sequences of partitions computed by Bovet et al. (2022) are available at: `https://dataverse.harvard.edu/file.xhtml?fileId=5657692`. We restricted the partitions to the $N = 281$ mice that were present throughout the full study period to ensure well-defined sequences of partitions, and considered the first nine temporal resolution values $\tau_i$, $i = 1, \ldots, 9$, since $\theta_{\tau_{10}}$ is an outlier. Note that the sequences tend to be fine-graining, see Fig. 13.

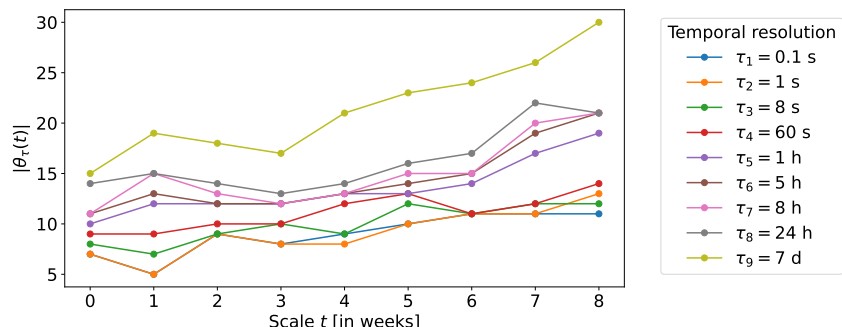

Figure 13: Number of clusters over weeks $t$ for different temporal resolutions $\tau$, where larger values of $\tau$ produce a higher number of clusters because of the increased temporal resolution.

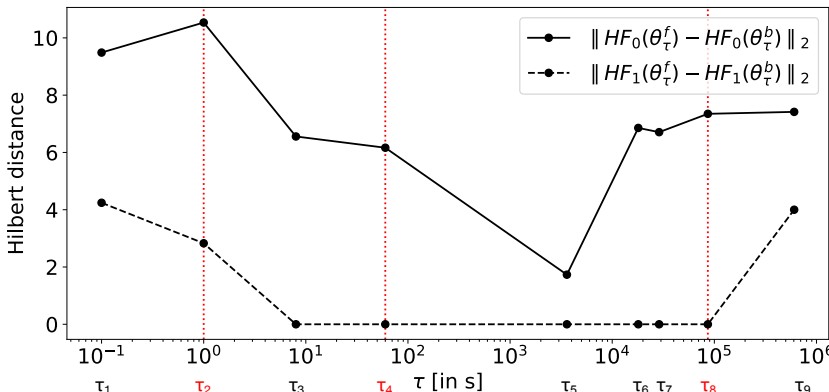

Figure 14: Hilbert distance between forward and backward sequences $\theta_\tau^f$ and $\theta_\tau^b$ for different temporal resolutions $\tau$.

**Time Reversibility.** In the main text, we restricted our analysis to the so-called *forward* Flow Stability sequences of partitions. However, by reversing time direction, Bovet et al. (2022) computed a second set of *backward* sequences. For each temporal resolution $\tau_i$, we thus get a forward and backward sequence denoted by $\theta_{\tau_i}^f$ and $\theta_{\tau_i}^b$, respectively. Here we use the MCbiF to compare the forward and backward sequences of partitions for different $\tau_i$, and we compute the Hilbert distance $\| HF_k(\theta_{\tau_i}^f) - HF_k(\theta_{\tau_i}^b) \|_2$ for $k = 0, 1$, see Fig. 14.

We observe that the Hilbert distance between forward and backward sequences is high for $\tau_2$ because the large-scale group structure changes significantly over the study period, so that the temporal

flows at low resolution $\tau_2$ are not reversible. In contrast, the Hilbert distance between forward and backward sequences is lower for $\tau_8$ because the underlying social groups are more stable over the study period, leading to increased time reversibility at the high temporal resolution. However, $\tau_4$ leads to the lowest Hilbert distance between forward and backward sequences, showing that the temporal resolution not only leads to the most hierarchical forward sequence but also to temporal flows that are most time reversible.

# E  BACKGROUND

## E.1  SANKEY DIAGRAMS

Non-hierarchical sequences of partitions $\theta$ are visualised by $M$-layered flow graphs $S(\theta) = (V = V_1 \uplus ... \uplus V_M, E = E_1 \uplus ... \uplus E_{M-1})$ called *Sankey diagrams* (Sankey, 1898; Zarate et al., 2018), where each level $m = 1, \ldots, M$ corresponds to a partition and vertices $V_m$ represent its clusters while the directed edges $E_m$ between levels indicate the overlap between clusters:

$$V_m := \{(m,i) \mid 1 \le i \le |\theta(t_m)|\}; \ E_m = \{[(m,i),(m+1,j)] \mid \theta(t_m)_i \cap \theta(t_{m+1})_j \neq \emptyset\}, \quad (9)$$

where $[u,v] \in E_m$ denotes a directed edge from $u \in V_m$ to $v \in V_{m+1}$. If $\theta$ is hierarchical, the Sankey diagram $S(\theta)$ is a directed tree—a merge-tree if $\theta$ is coarse-graining, or a split-tree if $\theta$ is fine-graining. The graph $S(\theta)$ is also called an *alluvial diagram* (Rosvall & Bergstrom, 2010).

Sankey diagrams are studied in computer graphics as they allow for the visualisation of complex relational data. In this context, a Sankey diagram is represented as a layout on the plane, whereby the nodes in each layer $V_m$ are vertically ordered according to a ranking $\tau_m : V_m \to \{1, \ldots, |V_m|\}$, and the layout of the Sankey diagram is then defined by the collection of such rankings, $\tau := (\tau_1, \ldots, \tau_M)$. For visualisation purposes, the layered layout should ideally minimise the number of crossings between consecutive layers, where a crossing between two edges $[u,v], [u',v'] \in E_m$ occurs if $\tau_m(u) > \tau_m(u')$ and $\tau_{m+1}(v) < \tau_{m+1}(v')$ or vice versa, and the crossing number (Warfield, 1977) is given by:

$$\kappa_\theta(\tau) := \sum_{m=1}^{M-1} \sum_{[u,v],[u',v'] \in E_m} \mathbb{1}_{\tau_m(u) > \tau_m(u') \wedge \tau_{m+1}(v) < \tau_{m+1}(v')}, \quad (10)$$

where $\mathbb{1}$ denotes the indicator function. The crossing number $\kappa_\theta(\tau)$ of the layout of the Sankey diagram $S(\theta)$ can be minimised by permuting the rankings in the layers, $\tau_m$, and we denote the *minimum crossing number for the layout* as:

$$\overline{\kappa}_\theta := \min_\tau \kappa_\theta(\tau). \quad (11)$$

This problem is known to be NP-complete (Garey & Johnson, 2006) and finding efficient optimisation algorithms is an active research area (Zarate et al., 2018; Li et al., 2025).

## E.2  MULTIPARAMETER PERSISTENT HOMOLOGY

*Multiparameter persistent homology* (MPH) is an extension of standard persistent homology to $n > 1$ parameters, first introduced by Carlsson & Zomorodian (2009). We present here basic definitions, see Carlsson & Zomorodian (2009); Carlsson et al. (2009); Botnan & Lesnick (2023) for details.

**Simplicial Complex.**  $K$ be a *simplicial complex* defined for the set $X$, such that $K \subseteq 2^X$ and $\tau \in K$ for $\forall \tau \subseteq \sigma \in K$. The elements of $\sigma \in K$ are called *simplices* and a $k$-dimensional simplex (or $k$-*simplex*) can be represented as $\sigma = [x_1, ..., x_{k+1}]$ where $x_1, \ldots, x_{k+1} \in X$ and we have fixed an arbitrary order on $X$. Note that $k = 0$ corresponds to vertices, $k = 1$ to edges, and $k = 2$ to triangles. We define the $k$-skeleton $K_k$ of $K$ as the union of its $n$-simplices for $n \le k$. We also define $\dim(K)$ as the largest dimension of any simplex in $K$.

**Multiparameter Filtration.**  Let us define the parameter space $(P, \le)$ as the product of $n \ge 1$ partially ordered sets $P = P_1 \times \cdots \times P_n$, i.e., $\boldsymbol{a} \le \boldsymbol{b}$ for $\boldsymbol{a}, \boldsymbol{b} \in P$ if and only if $\boldsymbol{a}_i \le \boldsymbol{b}_i$ in $P_i$ for $i = 1, \ldots, n$. A collection of subcomplexes $(K^{\boldsymbol{a}})_{\boldsymbol{a} \in \mathbb{R}^n}$ with $K = \bigcup_{\boldsymbol{a} \in \mathbb{R}^n} K^{\boldsymbol{a}}$ and inclusion

maps $\{i_{a,b} : K^a \to K^b\}_{a \leq b}$ that yield a commutative diagram is called a *multiparameter filtration* (or *bifiltration* for $n = 2$). We denote by $\mathrm{birth}(\sigma) \subseteq P$ the set of parameters, called *multigrades* (or *bigrades* for $n = 2$), at which simplex $\sigma \in K$ first appears in the filtration. For example, the *sublevel filtration* $K^a = \{\sigma \in K \mid f(\sigma) \leq a\}$ for a filtration function $f : K \to P$ maps each simplex $\sigma$ to a unique multigrade $f(\sigma)$, i.e., $|\mathrm{birth}(\sigma)| = 1$. A filtration is called *one-critical* if it is isomorphic to a sublevel filtration, and *multi-critical* otherwise.

**Multiparameter Persistent Homology.** Let $H_k$ for $k \in \{0, \dots, \dim(K)\}$ denote the $k$-dimensional *homology functor* with coefficients in a field (Hatcher, 2002), see Appendix E.3 for details. Then $H_k$ applied to the multiparameter filtration leads to a *multiparameter persistence module*, i.e., a collection of vector spaces $(H_k(K^a))_{a \in \mathbb{R}^n}$, which are the homology groups whose elements are the generators of $k$-dimensional non-bounding cycles, and linear maps $\{\imath_{a,b} := H_k(i_{a,b}) : H_k(K^a) \to H_k(K^b)\}_{a \leq b}$ that yield a commutative diagram called *multiparameter persistent homology* (MPH). For dimension $k = 0$, $H_k$ captures the number of disconnected components and for $k = 1$, the number of holes. Note that, for $n = 1$, we recover standard persistent homology (PH) (Edelsbrunner et al., 2002).

**Hilbert Function.** While barcodes are complete invariants of 1-parameter PH ($n = 1$), the more complicated algebraic structure of MPH ($n \geq 2$) does not allow for such simple invariants in general; hence, various non-complete invariants of the MPH are used in practice. We focus on the $k$-dimensional *Hilbert function* (Harrington et al., 2019; Botnan & Lesnick, 2023) defined as

$$\mathrm{HF}_k : P \to \mathbb{N}_0, \ \mathbf{a} \mapsto \mathrm{rank}[H_k(i_{a,a})] = \dim[H_k(K^a)], \tag{12}$$

which maps each filtration index $\mathbf{a}$ to the $k$-dimensional Betti number of the corresponding complex $K^{\mathbf{a}}$. We call the $k$-dimensional MPH *trivial* if $\mathrm{HF}_k = 0$. The *Hilbert distance* is then defined as the $L_2$ norm on the space of Hilbert functions and can be used to compare MPH modules.

### E.3 The Homology Functor

We provide additional background on simplicial homology and its functoriality, following Hatcher (2002).

**Simplicial Homology.** Let $K$ be a simplicial complex defined on the finite set $X$. For a fixed field $\mathbf{k}$ (the `RIVET` software uses the finite filed $\mathbf{k} = \mathbb{Z}_2$ (Wright & Zheng, 2020)) and for all dimensions $k \in \{0, 1, ..., \dim(K)\}$ we define the $\mathbf{k}$-vector space $C_k(K)$ whose elements $z$ are given by a formal sum

$$z = \sum_{\substack{\sigma \in K \\ \dim(\sigma) = k}} a_\sigma \sigma \tag{13}$$

with coefficients $a_\sigma \in \mathbf{k}$, called a $k$-*chain*. Note that the $k$-dimensional simplices $\sigma = [x_0, x_1, ..., x_k] \in K$ form a basis of $C_k(K)$. For a fixed total order on $X$, the *boundary operator* is the linear map $\partial_k : C_k \longrightarrow C_{k-1}$ defined through an alternating sum operation on the basis vectors $\sigma = [x_0, x_1, ..., x_k]$ given by

$$\partial_k(\sigma) = \sum_{i=0}^{k} (-1)^i [x_0, x_1, ..., \hat{x}_i, ..., x_k],$$

where $\hat{x}_i$ means that vertex $x_i$ is deleted from the simplex $\sigma$. The boundary operator fulfils the property $\mathrm{im}\,\partial_{k+1} \subset \ker \partial_k$. Hence, it connects the vector spaces $C_k$, $k \in \{0, 1, ..., \dim(K)\}$, through linear maps

$$\dots \xrightarrow{\partial_{k+1}} C_k \xrightarrow{\partial_k} C_{k-1} \xrightarrow{\partial_{k-1}} \dots \xrightarrow{\partial_2} C_1 \xrightarrow{\partial_1} C_0 \xrightarrow{\partial_0} 0,$$

leading to a sequence of vector spaces called *chain complex*. The elements in $Z_k := \ker \partial_k$ are called $k$-*cycles* and the elements in $B_k := \mathrm{im}\,\partial_{k+1}$ are called $k$-*boundaries*. Finally, the $k$-*th homology group* $H_k$ is defined as the quotient of vector spaces

$$H_k := Z_k / B_k, \tag{14}$$

whose elements are equivalence classes $[z]$ of $k$-cycles $z \in Z_k$. Each equivalence class $[z] \neq 0$ corresponds to a generator of non-bounding cycles, i.e., $k$-cycles that are not the $k$-boundaries of $k + 1$-dimensional simplices. This captures connected components at dimension $k = 0$, holes at $k = 1$ and voids at $k = 2$.

**Functoriality of $H_k$.** For fixed $k$, $H_k$ can be considered as a functor $H_k : \mathbf{Top} \to \mathbf{Vect}$, where $\mathbf{Top}$ denotes the category of topological spaces whose morphisms are continuous maps and $\mathbf{Vect}$ the category of vector spaces whose morphisms are linear maps. In particular, each topological space $K$ is sent to a vector space $H_k(K)$ and a continuous map $g : K \to K'$ is sent to a linear map $H_k(g) : H_k(K) \to H_k(K')$ such that compositions of morphisms are preserved, i.e., $H_k(g \circ f) = H_k(g) \circ H_k(f)$ for two continuous maps $f$ and $g$.

### E.4 Zigzag Persistence

We provide background on *zigzag persistence*, which was first introduced by Carlsson & de Silva (2010). For additional details, see Dey & Wang (2022).

**Zigzag Filtration.** Let $K^m$ be a simplicial complex defined on the set $X$ for $m = 1, \ldots, M$. If either $K^m \subseteq K^{m+1}$ or $K^{m+1} \subseteq K^m$, for all $m = 1, \ldots, M$, we call the following diagram a *zigzag filtration*:

$$K^1 \leftrightarrow K^2 \leftrightarrow \cdots \leftrightarrow K^{M-1} \leftrightarrow K^M,$$

where $K^m \leftrightarrow K^{m+1}$ is either a forward inclusion $K^m \hookrightarrow K^{m+1}$ or a backward inclusion $K^m \hookleftarrow K^{m+1}$. While forward inclusion corresponds to simplex addition, backward inclusion can be interpreted as simplex deletion.

**Zigzag Persistence.** Applying the homology functor $H_k$ to the zigzag filtration leads to a so called *zigzag persistence module* given by:

$$H_k(K^1) \leftrightarrow H_k(K^2) \leftrightarrow \cdots \leftrightarrow H_k(K^{M-1}) \leftrightarrow H_k(K^M),$$

where $H_k(K^m) \leftrightarrow H_k(K^{m+1})$ is either a forward or backward linear map. Using quiver theory, it can be shown that a zigzag persistence module has a unique interval decomposition that provides a barcode as a simple invariant.

## F  Details on Information-based Baseline Methods

Information-based measures can be used to compare arbitrary pairs of partitions in the sequence $\theta$ (Meilă, 2007). Assuming a uniform distribution on $X$, the conditional probability distribution of $\theta(t) = \{C_1, \ldots, C_n\}$ given $\theta(s) = \{C'_1, \ldots, C'_m\}$ is:

$$P_{t|s}[i|j] = \frac{|C_i \cap C'_j|}{|C'_j|}, \tag{15}$$

and the joint probability $P_{s,t}[i, j]$ is defined similarly. The conditional entropy (CE) $\mathrm{H}(t|s)$ is then given by the expected Shannon information:

$$\mathrm{H}(t|s) = -\sum_{i=1}^{|\theta(t)|} \sum_{j=1}^{|\theta(s)|} P_{s,t}[i,j] \log(P_{t|s}[i|j]) \tag{16}$$

It measures how much information about $\theta(t)$ we gain by knowing $\theta(s)$. If $\theta(s) \leq \theta(t)$ there is no information gain and $\mathrm{H}(t|s) = 0$. To summarise the pairwise conditional entropies in the sequence $\theta$, we define the $M \times M$ *conditional entropy matrix* CE by:

$$\mathrm{CE}_{i,j} := H(t_i|t_j), \tag{17}$$

for $i, j \in \{1, \ldots, M\}$. Furthermore, we can compute the variation of information (VI) $\mathrm{VI}(s,t) = \mathrm{H}(s|t) + \mathrm{H}(t|s)$, which is a metric. Both CE and VI are bounded by $\log N$.

Extending information-based measures for the analysis and comparison of more than two partitions is non-trivial. However, the pairwise comparisons can be summarised with the *consensus index* (Vinh et al., 2010) which can be computed as the average VI:

$$\overline{\mathrm{VI}}(\theta) := \frac{\sum_{i=1, i<j}^M \mathrm{VI}(t_i, t_j)}{M(M-1)/2} \tag{18}$$

