# OpenReview forum: "MCbiF: Measuring Topological Autocorrelation in Multiscale Clusterings via 2-Parameter Persistent Homology"
_ICLR.cc/2026/Conference — ICLR 2026 Poster_

### Official Review · Reviewer_5d9w · 2025-10-30

**Soundness:** 3
**Presentation:** 2
**Contribution:** 3
**Rating:** 6
**Confidence:** 3

**Summary:**

This paper proposes a method to analyze the defect if hierarchically in a
clustering using novel filtrations for multiparameter persistent homology.

The multipersistence module induced by this filtration is shown to satisfy
several desirable properties, such as being decomposable in
rectangles, or characterize defects in a nonhierarchical clustering.

**Strengths:**

- Detailed mathematical analysis. The proof seem sound.
  - The resulting filtration is mathematically satisfying.
  Rectangle decomposability a good to have.
  - The analysis with the 0,1-conficts is also interesting

**Weaknesses:**

- Code not available.
 - Maybe a bit math-heavy for this conference. I also think a little drawing
 could help for some definition/result.
 - Several definitions / proofs are hard to follow.
 - Experiments.
   - I'm not familiar with the state of the art, but from what I see, the only
   competitor is the conditional entropy introduced in 2003. It is thus hard
   to judge the practical impact of this construction.
   - I guess that this filtration can become easily very large but $N,M$ feel
   a bit small. How does this filtration scale w.r.t. these parameters? and
   what about its nerve equivalent?



minor comments.
- l157. I think there is an issue with the indices. shouldn't it be
$m,m,m+1,m+1$ ?
- l173. Shouldn't the $P_i$s be totally ordered? "operation of building
subsets" this is a bit vague.
- l216. Every multifiltration is multi-critical?
- l248. Unless I missed something, Rivet can handle arbitrary module
presentation, so MPH can be computed for arbitrary homological degree. Rivet is
also slightly outdated to compute the hilbert function. For instance, `mpfree`
can compute the betti numbers significantly faster (and for arbitrary degree as
well), for which there is a python interface in `multipers`. Furthermore, as
the induced module rectangle decomposable (Prop 21), the rectangle
decomposition (computable in `multipers`) should recover the module.
- Def 7. Not very intuitive.
- l269. What is a strictly hierarchical clustering? Is the relation strict on
the restriction on which it is non-constant?
- l303. $C$
- l333. $y$
- Table 1. Who is raw $\theta$, and what about corollary 15 instead of feeding
it to a regresser?
- prop 21. finitely presentable? and in the proof, I think it requires prop 23.
- l701. since $\theta$ is piecewise constant?
- prop 23. I think this should be detailed a bit more.

**Questions:**

see weaknesses.
 - Are there some stability result w.r.t. $\theta$?
 - Is there a link with the interlevelset filtration (which has similar
 guarantees)? I'm not sure this is possible since IIC, the Mayer-Vietoris
 sequence isn't valid for the non-nerve version, but this might be possible
 with the nerve version?

---

> ### Author Response · Authors · 2025-11-27
> **Response to Reviewer 5d9w (Part 1)**
>
> We thank the reviewer for the positive assessment of our work and the appreciation of our mathematical analysis. We are gratetful for the very detailed, insightful and constructive comments, which have helped us improve our presentation and sharpen some of our results.
>
> To address the issues raised,  we have provided a substantially revised version of our paper with an improved and extended presentation of all the theoretical results and additional clarifications and examples. Our main improvements are summarised in the general comment sent to all reviewers, but, more specifically, we addressed the following issues identified by the reviewer.
>
> - **Improved and extended presentation of mathematical theory and proofs:** We are grateful for the detailed comments by the reviewer that have helped us improve some of our definitions and proofs significantly. We have addressed all the issues identified, as detailed below. Additionally, we have improved the general presentation of our mathematical theory by making explicit the definitions of hierarchy, nestedness and the associated low-order and higher-order conflicts. This makes the presentation more rigorous and easier to understand. Accordingly, we also improved the statements of our definitions and propositions, as well as the presentation of our proofs, which are now organised by subsections in Appendix B. Finally, to give an overview of all the implications and relationships of our theory, we summarise our key theoretical results in Fig. 4 in Appendix A, where the mathematical links of all the results in the paper are presented consistently. We hope these improvements will be helpful and address the comments about the clarity of the presentation raised by the reviewer.
>
> - **Improved and additional visualisations:** As a first response to the request for improved visualisations, in the updated version, we have used third-party computational tools that compute optimised layouts for Sankey diagrams using non-crossing heuristics [1], and we have applied this tool to redraw all the Sankey diagrams in our paper. After doing so, the optimised visualisation layouts correspond more clearly to the topological patterns captured by the MCbiF. In fact, although the computational optimisation tool for Sankey layouts has been proposed as a greedy algorithm based on non-crossing heuristics, we show in our work that it is conceptually linked to our theoretical results (Corollary 19), and thus fits naturally with the mathematical setup of MCbiF. In fact, the direct connection between Sankey diagrams and MCbiF is now apparent in Fig. 3 (temporal social network of wild mice), where we see that the Sankey diagram of $\theta(\tau_8)$ presents a crossing that cannot be undone by relabelling the visualisation (see L484 - L487), as predicted by the presence of a 1-conflict from week 1 to 2, see Corollary 19. As a second response to the request for additional visualisations, we provide additional drawings for our different notions of conflicts in the improved Fig. 2 and also Fig. 7 in Appendix C. These serve to help visualise the patterns of interactions that lead to the topological conflicts in MCbiF. Furthermore, in Appendix C, we also added two new figures (Fig. 8 and 9) that illustrate the notion of 1-conflict in an additional, more complicated 4-element example (Example 38), and demonstrate why pairwise comparisons of partitions cannot capture 1-conflicts.
>
> - **Reproducibility:** The reproducibility of our experiments is crucial to us; hence, we will make code available on GitHub upon publication as already mentioned in the Reproducibility Statement of our original submission. Additionally, we are now preparing anonymised code to share with the reviewers and will alert them as soon as it becomes available.

---

> > ### Author Response · Authors · 2025-11-27
> > **Response to Reviewer 5d9w (Part 2)**
> >
> > - **Additional benchmarking against other measures:** In addition to conditional entropy (CE), we now benchmark our method against two further methods that are used to compare partitions (or clusterings): the classic Adjusted Rand Index (ARI) [2] and the recently proposed, state-of-the-art Maximum Overlap Distance (MOD) [3] method. Our numerical experiments confirm that the topological feature maps obtained from MCbiF outperform not only the CE but also the ARI and MOD feature maps on both regression and classification tasks in Sections 5.1 and 5.2. In the revised version, we have included these additional numerical results in Table 1 (and Table 3 in Appendix D.1) for the regression task, as well as in Table 2 (and Figure 12 in Appendix D.2)  for the classification task. Furthermore, we have also benchmarked our method against representation learning (with MLP and CNN) on the raw label encoding of $\theta$, see comment below. Tables 1 and 2  show that MLP and CNN models based on the MCbiF features outperform substantially models based on the raw label encodings (a direct representation of the data). We hope that these experiments help demonstrate the relevance of our method on various downstream machine learning tasks.
> >
> > - **Scalability of the method w.r.t. $N$ and $M$:** One of our drivers to explore theoretically the definition of the nerve-based MCbiF was the potential to improve the scalability of the method. For a sequence of $M$ partitions each defined on a set of $N$ elements,  Proposition 16 provides theoretical results about the dimensionality of the MCbiF and nerve-based MCbiF in terms of the parameters $M$ and the largest cluster size in $\theta$, which is bounded by $N$. The proposition shows that when $M$ is smaller than the size of the largest cluster, then the nerve-based MCbiF is computationally advantageous. Our numerical experiments exploit this computational advantage (see next point).
> >
> > - **Choices of $N$ and $M$ in experiments:** The size of the space of partitions defined on $N$ elements grows exponentially according to the Bell number $B_N$, which is already very large for $N=10$ ($B_{10}=115,975$). Moreover, the space of coarse-graining sequences of length $M$ and partitions on $N$ elements grows much faster than the Bell numbers, i.e., $|\Pi_{N}^{M}| \gg B_{N}$. Hence, we chose a relatively small $N=5, 10$ and $M=20$ in the regression task (Section 5.1), so that a sample of 20,000 sequences leads to a representative distribution of the minimal crossing number of the corresponding Sankey diagrams. To demonstrate the scalability of our method, we chose larger $N=500$ and $M=30$ for the classification task (Section 5.2). Furthermore, our real-world application is to a medium-sized social network temporal dataset, which features sequences of partitions defined on $N=281$ elements (corresponding to the different mice) over $M=9$  weeks. In this example, we found that the nerve-based MCbiF induces a 50-fold reduction in computation time due to a much lower number of simplices (260 simplices for the nerve-based MCbiF vs. 116,700 for the original MCbiF). This is consistent with our theoretical results about the dimensionality of the MCbiF and nerve-based MCbiF (in Proposition 16).
> >
> >
> > **Minor issues:** Thank you for raising all these minor issues, which we have addressed as follows:
> >
> > - **L157 (now L1451):** We fixed the indices.
> >
> > - **L173 (now L1478):** The $P_i$ are totally ordered sets in our case, but in general they can be partially ordered, see reference [4, p. 7].
> >
> > - **L216 (now L170):** We follow the definitions of multi-critical and 1-critical from reference [4, Def. 2.2], where it is emphasised that not every multiparameter filtration is multi-critical. For example, any sublevel filtration defined on a poset $P=P_1\times P_2$ is 1-critical, see [4, Def. 2.2]. In contrast, the MCbiF is multi-critical because it is not isomorphic to a sublevel filtration, see the proof of Proposition 4. In contrast, the nerve-based MCbiF is 1-critical because every simplex has a unique bigrade at which it is born (see also our reply to your question about the interlevel set filtration below).

---

> > > ### Author Response · Authors · 2025-11-27
> > > **Response to Reviewer 5d9w (Part 3)**
> > >
> > > - **L248 (now L193):** Thank you very much for this comment. We have carefully checked the ``RIVET`` Python API and realised that (as pointed out by the reviewer) one can indeed compute MPH for arbitrary homology dimension (also called homology degrees). We thank you for pointing out this error on our side, which we have corrected in the updated version. Therefore, one can also define the notion of $k$-conflicts for $k\ge 2$ following the results in [5, Remark 19]. However, we leave the analysis of $k$-conflicts with MCbiF for future work due to the length and time  constraints of this conference paper. Thanks also for pointing us to the ``multipers`` software. It seems like a very useful piece of software that we would like to use in future work when exploring richer algebraic invariants of the MCbiF beyond Hilbert functions.
> > >
> > > - **Def. 7 (now Def. 5):** We thank the reviewer for highlighting that our original definition of a 0-conflict was not very intuitive. We have improved our definition and now say that $\theta$ has a 0-conflict in $[s,t]$ when there is no maximum in the subposet $\theta[s,t]$. This is now the much clearer Def. 5 and an illustration can be found in Fig. 2b. (Note that we arrived at this definition by using what had been called property C2 in Prop. 8 of the original submission.) Additionally, we now make clear the notion of a triangle 0-conflict, illustrated in Fig. 2c. In Prop. 6 and 7, we link our notions of 0-conflict and triangle 0-conflict to non-nestedness and non-hierarchy in $\theta$. Furthermore, we characterise triangle 0-conflicts further by analysing the graph-theoretic properties of the 1-skeleton of the MCbiF in Prop. 30 in Appendix B.2.1. We summarise our theoretical results about the relationships between conflicts, non-hierarchy and Hilbert functions in Fig. 4 in Appendix A. We hope that these improvements and additions make our statements clearer.
> > >
> > > - **L269 (now L234):** We have introduced a formal definition of hierarchy and strict hierarchy in lines L063 to L066. By strictly hierarchical, we mean that the sequence of partitions is hierarchical over all scales.
> > >
> > > - **L303 (now L291):** We fixed the typo; it should be $C$.
> > >
> > > - **L333 (now L338):** We fixed the typo, with thanks.
> > >
> > > - **Table 1:** Thanks for this question. We realise that our original version was not clear in explaining the meaning of “raw $\theta$”. We have expanded this explanation in the current version (L375-L377). This is now renamed as “Raw label encoding” of $\theta$, and it corresponds to a (non-unique) $N \times M$ matrix containing the labels of the clusters of the set of $N$ elements for the $M$ scales in the sequence of partitions $\theta$. This is then treated as a representation of $\theta$ and used as an input for representation learning models (MLP, CNN, etc) for both the regression and the classification tasks. We hope this clarifies this issue.
> > >
> > > - **Using our lower bound for the minimal crossing number as a regressor:** We appreciate this interesting suggestion by the reviewer and have evaluated if our lower bound on the minimal crossing number based on the off-diagonal elements of the HF1 (Corollary 19) performs well in the regression task (Section 5.1). We found that the performance of the lower bound to predict the actual minimal crossing number is actually very low: $R^2=-3$ for $N=5$ and $R^2=-31$ for $N=10$. Hence, we have not included these results in Table 1. This result is not surprising because, as already discussed in Remark 20, the lower bound only captures crossings that arise from one layer to the next and we hypothesise there that the full HF0 and HF1 feature maps capture more complicated crossings that arise in the Sankey layout across many layers.  If the reviewer deems this result interesting, we will be happy to add it to the paper.
> > >
> > > - **Prop. 21 (now Prop. 27):** We now use the term “finitely presented” consistently in our paper, following reference [4, p. 35]. As pointed out by the reviewer, the proof of the proposition requires Prop. 23 (now Prop. 34). We make this explicit in our improved presentation of our proof (L857).  Thanks for pointing out this link which helps clarify our presentation.
> > >
> > > - **L701 (now L869):** Exactly; as $\theta$ is a piecewise-constant function with change points $t_1<\dots<t_M$, we can assume in the proof of Prop. 27 (previously Prop. 21) without loss of generality that $t=t_k$, $t'=t_\ell$, $t''=t_m$, $t'''=t_n$ for change points $t_k<t_\ell<t_m<t_n$ of $\theta$.

---

> > > > ### Author Response · Authors · 2025-11-27
> > > > **Response to Reviewer 5d9w (Part 4)**
> > > >
> > > > - **Prop. 23 (now Prop. 34):** Thanks for pointing out that additional explanations are useful for this proof. The proof of the proposition relies on a persistent nerve lemma for abstract simplicial complexes from [5, Lemma 32]. We have now included this result as Lemma 33 in Appendix B.2.2. In the proof of the proposition (now Prop. 34), we now explain in detail how the Lemma can be used to prove the equivalence between the MPH of the MCbiF and the nerve-based MCbiF, see L1025 to L1036.
> > > >
> > > > We believe that addressing all the issues pointed out by the reviewer has helped us to improve significantly the presentation of our mathematical content, and we would like to thank the reviewer again for the detailed comments.
> > > >
> > > > **Questions raised by the reviewer:**  We thank the reviewer for these two interesting questions. These deserve to be the object of future work, but we provide some initial replies to them here:
> > > >
> > > > - There are already stability results w.r.t. $\theta$ established for the 1-parameter MCF filtration [5, Remark 9 & Prop. 11], which corresponds to the top row in the MCbiF diagram (see Remark 24 in Appendix B.2). We agree with the reviewer that it would be important to extend these stability results to the 2-parameter MCbiF. Due to the constraints of this conference paper, we plan to address this research in future work.
> > > >
> > > > - Regarding the second question, the reviewer is right in assuming that the MCbiF is generally not an *interlevel set filtration* as defined in [6, p. 3136] because it is multi-critical according to Prop. 4. However, as the nerve-based MCbiF is 1-critical instead, it could be understood as an interlevel set filtration w.r.t. a filtration function $\gamma$ that maps each simplex (corresponding to cluster indices of intersecting clusters) to the smallest pair of scales $(s,t)$ (according to the partial order $\mathbb{R}^{op}\times\mathbb{R}$) such that all the clusters are present in partitions $\theta([s,t])$. Investigating further the topological properties of the filtration function $\gamma$ will also be the focus of future work.
> > > >
> > > >
> > > > **References**
> > > >
> > > > [1] Li, S. et al. OmicsSankey: Crossing Reduction of Sankey Diagram on Omics Data. 2025.06.13.659656 Preprint at https://doi.org/10.1101/2025.06.13.659656 (2025).
> > > >
> > > > [2] Hubert, L. & Arabie, P. Comparing partitions. Journal of Classification 2, 193–218 (1985).
> > > >
> > > > [3] Peixoto, T. P. Revealing Consensus and Dissensus between Network Partitions. Phys. Rev. X 11, 021003 (2021).
> > > >
> > > > [4] Botnan, M. B. & Lesnick, M. An introduction to multiparameter persistence. in EMS Series of Congress Reports (eds Buan, A. B., Krause, H. & Solberg, Ø.) vol. 19 77–150 (EMS Press, 2023).
> > > >
> > > > [5] Schindler, J. & Barahona, M. Analysing Multiscale Clusterings with Persistent Homology. Preprint at https://doi.org/10.48550/arXiv.2305.04281 (2025).
> > > >
> > > > [6] Botnan, M. & Lesnick, M. Algebraic stability of zigzag persistence modules. Algebraic & Geometric Topology 18, 3133–3204 (2018).

---

> > > > > ### Author Response · Authors · 2025-11-28
> > > > > **Release of Anonymised Code To Reproduce Numerical Experiments**
> > > > >
> > > > > We updated the Reproducibility Statement and now provide an anonymous repository with a Python implementation of MCbiF and scripts to reproduce all numerical experiments and figures: https://anonymous.4open.science/r/mcbif-iclr-468A/. This resolves the noted weakness—code availability—and improves reproducibility and ease of use for other researchers.

---

### Official Review · Reviewer_tfaN · 2025-10-31

**Soundness:** 4
**Presentation:** 4
**Contribution:** 4
**Rating:** 8
**Confidence:** 4

**Summary:**

The paper considers the following setting. Given is finite set $X$. For each real number $t \geq 0$, there is a partitioning of $X$, in other words, we have a family of different clustering ways. Importantly, it is not assumed to be hierarchical. The paper proposes a way to use 2-parameter persistent homology to describe such a family. The construction is elegant: given values $s \leq t$, one looks at all the clusters occurring between $s$ and $t$ and inserts a corresponding simplex into the complex $K_{s,t}$.
The paper proves algebraic properties of the resulting 2-persistence modules (pointwise finite-dimensional, finitely presentable and block-decomposable). Then the paper concentrates on the Hilbert functions (just pointwise dimensions) of the resulting bifiltrations and derives new measures of conflict and interpretations of the information captured by the TDA techniques. There are also experiments, offering new insights for, e.g., a real-world dataset showing how the grouping of mice changes over time.

**Strengths:**

- The construction of bifiltration is really nice and elegant. While the formal definition might be hard to digest for non-experts, the illustration in Fig. 1 does a great job explaining what is really going on.
- The paper provides new quantitative measures to clustering.
- The results generalize the previously known Sankey diagrams.

**Weaknesses:**

- As the paper admits, Hilbert functions are a rather crude invariant that loses some information. The paper poses exploration of other invariants as future work, which is fair enough.
- The first experiment (predicting crossing number) seems to be a bit weak, considering that even for MCBif the Pearson is still 0.544. I mean, the whole set up looks a bit artificial, but I am not an expert in clustering.

**Questions:**

I wonder if every block-decomposable module can be obtained from some clustering by the proposed construction? Of course, the module must also be assumed to be zero outside of the $s \leq t$ triangle.

---

> ### Author Response · Authors · 2025-11-26
> **Response to Reviewer tfaN (Part 1)**
>
> We thank the reviewer for the positive assessment of our work and the appreciation of the mathematical content and numerical experiments. As described in the general comment sent to all reviewers, we have uploaded a revised version of the paper that contains a number of substantial improvements, which further strengthen our paper and clarify different aspects of the theory, as well as additional examples that illustrate the findings. Here we address some of the specific comments raised in your review.
>
> **Topological invariants of MCbiF and additional theoretical results:** As noted in the original submission (and mentioned by the reviewer), we decided to focus on Hilbert functions as invariants due to their analytical simplicity, which allowed us to prove a range of theoretical results.  In the updated version, we have clarified and expanded the theoretical links between the topology of MCbiF with hierarchy, nestedness and higher-order inconsistencies (conflicts) in sequences of partitions (Section 3.2). Additionally, we have extended the mathematical links of the nerve-based MCbiF with Sankey diagrams (and their crossings) and some of the graph-theoretical properties of the Sankey diagram (Propositions 17 and 18). Our analytical framework also allows us to examine limiting cases where the advantages of the higher-order inconsistencies detected by MCbiF are distinct from the pairwise cluster comparisons based on, e.g., the information-based conditional entropy (CE), see Example 39 in Appendix C. Despite their simplicity, Hilbert functions still yield good performance in our numerical experiments (see Tables 1 and 2). We plan to use more advanced algebraic invariants of the MCbiF persistence module in future work, e.g., using the Multipers software suggested by reviewer 5d9w.
>
> **Clarification and extension of the theoretical results:**  In the new version, we have made a thorough effort in rewriting and simplifying statements and proofs. This includes the clarification of the relationship of the Hilbert functions with nestedness and hierarchy and the ensuing conflicts, as well as direct mathematical links between MCbiF and the multilayer graph underpinning Sankey diagrams. The patterns associated with the conflicts have also been made more transparent by adding some additional illustrations. Appendices A and B contain additional explanations of key concepts, and the proofs are organised by section. We have also added new examples in Appendix C to clarify the emergence and resolution of the different types of conflicts. We also explain with examples how the Hilbert functions of the MCbiF include (and supersede) the information contained in dendrograms and in information-theoretic measures of cluster comparison. Finally, as an overall summary, we also provide a figure that brings together all the theoretical results and their mathematical relationships (Figure 4 in Appendix A). We hope that all these improvements will help clarify further our contributions.
>
> **Regression task:** The reviewer comments on the performance on the regression task of predicting the crossing number. Our bifiltration provides a general description of the sequence of partitions as a whole, and is not fine-tuned for any specific task. In fact, the (combinatorial) optimisation of the Sankey layout is NP-hard and hence provides a computationally demanding regression task starting from features of the sequence of partitions (without any algorithmic dynamic optimisation). This is shown by the poor performance of all models based on different sets of features (in some cases with correlations close to zero).  Hence, we view the 0.54 correlation achieved by using our MCbiF invariants as (relatively) good performance. To further emphasise the connection between our topological features and Sankey diagrams, all the Sankey diagram layouts in the updated version have been redrawn after being optimised using the OmicsSankey package that implements a greedy optimisation that minimises crossings of the layout.  The examination of this issue in more detail has also unveiled further theoretical connections of this computational problem. In Corollary 19, we proved already in our initial submission that there is a link between our 1-dimensional Hilbert function and the minimum crossing number in the Sankey layout. We have clarified this result further in the updated version as a further connection of our results with difficult optimisation problems.
>
> **References:**
>
> [1] Li, S. et al. OmicsSankey: Crossing Reduction of Sankey Diagram on Omics Data. 2025.06.13.659656 Preprint at https://doi.org/10.1101/2025.06.13.659656 (2025).

---

> > ### Author Response · Authors · 2025-11-26
> > **Response to Reviewer tfaN (Part 2)**
> >
> > **Regarding your question:** The theoretical question you posed (i.e., whether every block-decomposable module could be obtained via the MCbiF of a sequence of partitions) is very interesting. We would like to dedicate more time to this question in future work, but let us present here some preliminary thoughts.
> >
> > We believe that it is not possible to find for every block-decomposable module a sequence of partitions such that the homology functor of a specified dimension applied to the MCbiF leads to the module. To see this, consider the simple module defined on a single-element parameter set $P=\{ (0,0) \}$ such that $(0,0)\mapsto \mathbb{Z}$. If we require dimension $k=1$, then there is no sequence of partitions with length $M=1$, i.e., a single partition, such that the 1-dimensional homology of the MCbiF is equal to that module because the clusters in any single partition are mutually exclusive and so the simplicial complex $K^{0,0}$ consists of mutually disjoint solid simplices that have trivial 1-dimensional homology. If we have no requirement on the dimension, we could, of course, choose a single partition on the 1-element set so that the 0-dimensional homology of the MCbiF would lead to the module defined above.
> >
> > This example shows that extending MCbiF to sequences of soft partitions, i.e., partitions with overlapping clusters, could be an interesting avenue of future research. The question of whether every block-decomposable module could be obtained via the MCbiF of a sequence of soft partitions would then arise as another intriguing research direction.

---

### Official Review · Reviewer_NmJi · 2025-11-01

**Soundness:** 2
**Presentation:** 2
**Contribution:** 2
**Rating:** 2
**Confidence:** 3

**Summary:**

This paper introduces the Multiscale Clustering Bifiltration (MCbiF), a topological framework for analyzing multi-resolution, non-hierarchical clusterings that describe data at different levels of coarseness. MCbiF models how clusters intersect and evolve across scales using a two-parameter filtration of simplicial complexes, whose structure is studied through multiparameter persistent homology. The resulting stable Hilbert functions capture key topological properties: dimension-0 features measure the nestedness of clusters, while dimension-1 features quantify higher-order inconsistencies between them. The method generalizes dendrograms (for hierarchical cases) and extends Sankey diagrams to represent higher-order relationships.

**Strengths:**

The paper is well-grounded in theory and method, with clear mathematical development and interpretable ideas. Its main strength lies in the clarity and rigor of the framework, which extends topological concepts in a coherent way to analyze multiscale clustering.

**Weaknesses:**

However, the empirical evaluation is limited and lacks modern benchmarking. Experiments are mostly conducted on synthetic datasets and a single small-scale real-world example (wild mice social groups), which limits generalizability. The comparative analysis is restricted to information-theoretic baselines (conditional entropy), with no inclusion of state-of-the-art representation learning methods such as graph neural networks or topological embeddings. Performance variability and robustness analyses are not reported, and code availability is not mentioned, hindering reproducibility. While the theoretical development is strong, the translation to practical machine learning applications is limited, reducing the paper's accessibility and impact for a broader representation learning audience.

**Questions:**

- Could the authors elaborate on how MCbiF could be applied to modern large-scale or high-dimensional datasets (e.g., image, text, or graph data)? This would help clarify the method's practical relevance beyond theoretical or synthetic settings.

- Why were only information-theoretic measures (e.g., conditional entropy) used as baselines? Including comparisons with recent representation learning or topological deep learning methods (e.g., graph neural networks, persistence-based embeddings) could strengthen the empirical validation.

- While the paper provides theoretical insight into Hilbert functions, could the authors give more intuitive examples or visualizations showing how specific topological patterns correspond to interpretable data behaviors?

---

> ### Author Response · Authors · 2025-11-26
> **Response to Reviewer NmJi (Part 1)**
>
> We thank the reviewer for the comments raised. To address them, we have improved our manuscript significantly, as outlined in the general comment above. In particular, our revised version addresses the following questions raised by the reviewer:
>
> **Applicability:** The MCbiF is designed for the analysis of non-hierarchical sequences of partitions, an important data type in modern data analysis whereby multiscale representations are obtained from unsupervised clustering methods. Such data are pervasive across applications, from imaging to text, from biology to engineering or social networks. Our real-world application to the analysis of the temporal evolution of a social network of mice illustrates this.  We now emphasise these areas of wide applicability explicitly in our updated Introduction, where we have included additional references and examples.
>
> However, this is a data form that has been largely inaccessible to current machine learning methods. Indeed, analysing and comparing non-hierarchical sequences of partitions has received little attention in the machine learning literature to date. When sequences of partitions are hierarchical, they correspond to acyclic merge trees (also called dendrograms), and there is important literature on this problem linking these datasets to ultrametrics. Yet such methods are not applicable to non-hierarchical sequences (as we discuss in detail in our manuscript). Hence, multiscale, non-hierarchical partitions are usually characterised heuristically or partially, or are just visualised using Sankey diagrams. MCbiF fills this void by providing a systematic framework,  with a rigorous foundation in mathematical concepts, which can be applied to any hierarchical or non-hierarchical sequence of partitions across scales. Although our paper focuses on setting the mathematical foundations, our numerical examples have been chosen to showcase some of the use cases, spanning regression and classification tasks, as well as an unsupervised examination of a real-world dataset.  To give an indication of scalability, the classification task in Section 5.2 has been carried out on a relatively large dataset of 3,700 sequences of partitions of N=500 elements with M=30 scales. The nerve-based construction allows for the treatment of such cases without any computational optimisation on personal computers.
>
> In our updated version, we now emphasise all these points in our introduction and throughout the applications and the discussion.
>
> **Comparison to additional baseline feature maps:** In addition to conditional entropy (CE), we now benchmark our method against two further methods that are used to compare partitions (or clusterings): the classic Adjusted Rand Index (ARI) [1] and the recently proposed, state-of-the-art Maximum Overlap Distance (MOD) [2] method. Our numerical experiments confirm that the topological feature maps obtained from MCbiF outperform not only the CE but also the ARI and MOD feature maps on both regression and classification tasks in Sections 5.1 and 5.2. In the revised version, we have included these additional numerical results in Table 1 (and Table 3 in Appendix D.1) for the regression task, as well as in Table 2 (and Figure 12 in Appendix D.2)  for the classification task.
>
> **Comparison to representation learning:** We agree with the reviewer about the importance of comparing our MCbiF embeddings to representation learning methods. In fact, we had already done this in our initial submission by developing MLP and CNN models based on the raw label encodings of $\theta$ (i.e., the $N \times M$ matrix that contains the cluster labels of the $N$ elements over the $M$ partitions across scales). Tables 1 and 2  show that MLP and CNN models based on the MCbiF features outperform substantially models based on the raw label encodings (a direct representation of the data).
>
> We realise that our use of this raw data representation had not been made clear in our previous submission (in fact, reviewer 5d9w also asked about the meaning of what we previously called “raw $\theta$”). In our revised version, we explain in detail the meaning of the raw label encoding, and we frame the MLP and CNN models trained on the raw label encoding as learning representations of the data from scratch, yet with worse performance than models based on our topological MCbiF embeddings.  We will endeavour to add additional representation models before the deadline.

---

> > ### Author Response · Authors · 2025-11-26
> > **Response to Reviewer NmJi (Part 2)**
> >
> > **Interpretation of Hilbert functions:** In response to the referee’s comment requesting more intuitive clarification of the meaning of the Hilbert functions, we have expanded both examples and visualisations in our updated manuscript.
> >
> > Firstly, we have regenerated all the Sankey diagrams in the paper using a computational tool [3] that uses a non-crossing heuristic to optimise the Sankey layouts. This results in visualisation layouts that reveal more clearly the topological patterns captured by the MCbiF. Interestingly, this external layout optimisation tool [3] uses a heuristic that can be linked to our Hilbert function invariants and how they reveal crossing patterns and can also be seen as generalisations of Sankey diagrams. This whole set of connections has been expanded and made explicit in Proposition 18, Corollary 19, and Remark 20 in the updated paper and is also exploited in our regression task in Section 5.1. In fact, the connection between Sankey diagrams and MCbiF is now apparent in Fig. 3 (temporal social network of wild mice), where we see that the Sankey diagram of $\theta(\tau_8)$ presents a crossing that cannot be undone by relabelling the visualisation, as predicted by Corollary 19.
> >
> > Secondly, we have added a series of examples and figures that present the meaning of the Hilbert functions more intuitively. This includes the following: (i)  a figure where the different conflicts are presented in a Sankey diagram with one layer (Figure 2); (ii) an example and Sankey diagrams that illustrate the differences of what $\text{HF}_1$ captures versus the information-theoretic CE (Example 39); (iii) an example with a set of 4 elements that clearly shows the resolution of conflicts that extend across scales and thus cannot be detected by pairwise measures such as CE, ARI or MDO (Example 38).
> >
> >
> > **Reproducibility:** As mentioned in the Reproducibility Statement of our original submission, we will make code available on GitHub upon publication (this is always a strong commitment in our research practice). In fact, we are preparing anonymised code to share with all reviewers and will send an update to this comment as soon as it is available.
> >
> > **References:**
> >
> > [1] Hubert, L. & Arabie, P. Comparing partitions. Journal of Classification 2, 193–218 (1985).
> >
> > [2] Peixoto, T. P. Revealing Consensus and Dissensus between Network Partitions. Phys. Rev. X 11, 021003 (2021).
> >
> > [3] Li, S. et al. OmicsSankey: Crossing Reduction of Sankey Diagram on Omics Data. 2025.06.13.659656 Preprint at https://doi.org/10.1101/2025.06.13.659656 (2025).

---

> > > ### Author Response · Authors · 2025-11-28
> > > **Release of Anonymised Code To Reproduce Numerical Experiments**
> > >
> > > We have updated the Reproducibility Statement in our manuscript and now provide an anonymous, public repository with a Python implementation of MCbiF and scripts to reproduce all numerical experiments and figures: https://anonymous.4open.science/r/mcbif-iclr-468A/. Making the code open access facilitates peer-review and reproducibility of our results, addressing the concerns raised by the reviewer.

---

> > > > ### Author Response · Authors · 2025-12-03
> > > > **Update on Numerical Experiments**
> > > >
> > > > As suggested by the reviewer, we updated our manuscript with additional numerical results on representation learning with graph convolutional networks (GCN) for the regression task and conducted a performance variability analysis for both regression and classification tasks. We have detailed our updates in the general response above and summarise them here briefly:
> > > >
> > > > - **Representation learning with GCNs:** We thank the reviewer for suggesting to benchmark our method against state-of-the-art graph neural networks. As our target in the regression task (Section 5.1) is the minimal crossing number computed from the Sankey diagram layout, benchmarking against a graph convolutional network (GCN) trained on the Sankey graphs is indeed a natural baseline. We performed extensive hyperparameter optimisation for the GCN using the train and validation split of our data and evaluated the final performance on the test split (see Appendix D.1). We added our results to Tables 1 and 4 and found that the GCN performs significantly worse than models trained on the MCbiF feature maps. This complements the results on representation learning on the (non-unique) raw label encodings of $\theta$ with MLP and CNN performed previously. While GCN on Sankey graphs leads to higher performance than MLP and CNN on the raw label encodings as expected, all representation learning models are still outperformed by models trained on our MCbiF feature maps by a large margin. We confirmed this through our performance variability analysis (see comment below) and statistical tests on the test residuals.
> > > > - **Performance variability analysis:** We thank the reviewer for suggesting a performance variability analysis. To address this request, we computed bootstrapped 95% confidence intervals (CI) of the test $R^2$ score in the regression task (Section 5.1) and test accuracy in the classification task (Section 5.2). We present our results in Tables 3 and 5 in the Appendix. Our analysis shows that MCbiF consistently outperforms the other methods. In particular, for the regression task the best MCbiF model (trained on combined HF0 and HF1 features) leads to a CI whose lower bound on the test $R^2$ score is higher than the upper bound of the CIs of the other methods. The same consistently improved performance of the models trained on HF1 is observed for  the classification task.
> > > >
> > > > We believe that our additional numerical experiments and statistical analysis of performance variability complement, significantly strengthen the empirical validation of our method as asked for by the reviewer. For full reproducibility, code for our method and all numerical experiments (including our latest additions) is available in our anonymous repository: https://anonymous.4open.science/r/mcbif-iclr-468A/ We hope that our manuscript is now deemed publishable for ICLR.

---

### Author Response · Authors · 2025-11-25
**Revised version of our paper with extended improvements and additions - Comment 1**

We thank all reviewers for their constructive feedback. To address the comments of the reviewers, we provide a revised version of our paper with extended improvements and additions. Our main updates include:

- **Improved presentation of theory:** We have improved the presentation of our theory by making explicit the definitions of hierarchy, nestedness and the associated low-order and higher-order conflicts. This makes the presentation more rigorous and easier to understand. Accordingly, we also improved the statements of our definitions and propositions, as well as the presentation of our proofs, which are now organised by subsections in the Appendix. Finally, to give an overview of all the implications and relationships of our theory, we summarise our key theoretical results in Fig. 4 in Appendix A, where the mathematical links of all the results in the paper are presented consistently. We hope these improvements will be helpful and address the comments of the reviewers.
- **Improved visualisations:** Standard visualisation tools for Sankey diagrams are usually non-optimised, and those were used in the original version. In the updated version, we have used additional computational tools that compute optimised layouts for Sankey diagrams using non-crossing heuristics [1]. We have applied this tool to all the examples in our paper. In doing so, the optimised visualisation layouts correspond more clearly to the topological patterns captured by the MCbiF. In fact, the idea of using layout optimisation heuristics emerged from the realisation that some of our rigorous invariants are linked to crossing patterns in particular cases of Sankey diagrams. This is now discussed in Proposition 18, Corollary 19 and Remark 20 in our paper and exploited in our regression task in Section 5.1.
- **Additional illustrative examples:** We present expanded explanations and illustrations of examples illustrating different types of conflicts, special cases, links to Sankey diagrams (see Fig. 1 and Fig. 7 corresponding to Example 37 in Appendix C; Fig. 2; as well as Fig. 9 corresponding to Example 39 in Appendix C), plus an additional example illustrating a sequence of partitions for a 4-element set which highlights the emergence of more complex conflicts and their resolution  (Example 38 and Fig. 8 in Appendix C). All these examples serve to illustrate patterns of cluster interaction and the different notions of conflict, as well as providing visual examples for some of the propositions throughout the paper (properly referenced throughout). We hope these examples help clarify some of the comments raised by the reviewers.
- **Benchmarking against baseline feature maps:** A comment was raised about expanding our range of comparisons against other baseline feature maps in addition to conditional entropy (CE).  In response to this comment, we have benchmarked our method against two further methods for cluster comparisons: the popular Adjusted Rand Index (ARI) [2], which is another standard in the field, and the recently proposed,  state-of-the-art Maximum Overlap Distance (MOD) [3] method. Our computational results in Sections 5.1 and 5.2 show that the feature maps generated from  MCbiF (i.e., its Hilbert functions of dimensions 0 and 1)  outperform not only the CE but also the ARI and MOD feature maps on both regression and classification tasks. We have included the additional results in Tables 1 and 2 of the revised paper.
- **Comparison to representation learning:** There was a comment on comparing the performance of our ML models trained on MCbiF feature maps to models trained on the raw label encoding, a matrix containing the labels of the clusters for all the scales in the sequence of partitions $\theta$. To make this comparison clearer,  we show that representation learning on the raw label encodings leads to reduced performance when compared to representation learning on the MCbiF feature maps, see Tables 1 and 2 of the revised paper. This demonstrates the advantages of the representation obtained by using MCbiF to produce feature maps that capture topological invariants of the non-hierarchical sequences of partitions for downstream machine learning tasks.
- **Minor corrections:** We have corrected the minor issues identified by reviewer 5d9w, see the direct comment to the reviewer below.

We continue our response in the next comment below.

**References**:

[1] Li, S. et al. OmicsSankey: Crossing Reduction of Sankey Diagram on Omics Data. Preprint at https://doi.org/10.1101/2025.06.13.659656 (2025).

[2] Hubert, L. & Arabie, P. Comparing partitions. Journal of Classification 2, 193–218 (1985).

[3] Peixoto, T. P. Revealing Consensus and Dissensus between Network Partitions. Phys. Rev. X 11, 021003 (2021).

---

> ### Author Response · Authors · 2025-11-25
> **Revised version of our paper with extended improvements and additions - Comment 2**
>
> We continue our list of updates from the previous comment:
>
> - **Improved writing:** We improved the writing of our paper throughout to enhance clarity. We also improved the introduction to emphasise the need for analysing non-hierarchical sequences of partitions that arise from many applications, as suggested by the reviewers.
> - **Reproducibility:** The reproducibility of our experiments is crucial to us and central to our way of operating.  As already mentioned in the Reproducibility Statement of our original submission, we will make code available on GitHub upon publication. Additionally, we are now preparing anonymised code to share with the reviewers and will alert them as soon as it becomes available.
> - **Appendix material:** The Appendix has been restructured to facilitate easier access to the extensive support material, should the reviewers wish to use it. The Appendix now includes sections as follows:
>    - **Summary of key theory results:** Summary figure showing all the theoretical interconnections of our results.
>    - **Proofs:** Proofs to all our propositions in the main text, organised by section with clear links to the main body. It also contains additional supporting propositions, lemmas and corollaries (with proofs) that were not so central to the main arguments of the paper.
>    - **Examples:** detailed explanation of examples with figures and new examples highlighting the different notions of conflicts and the meaning of the Hilbert functions
>    - **Experiments:** Detailed explanation of datasets and computational setups for all our numerical experiments.
>    - **Background:** Extensive introduction to several of the theoretical topics for the interested reader, providing definitions, formalisation and some explanations. Note that to make space for our extended improvements and additions in the main text, we have moved the background and formal statements about Sankey diagrams and Multiparameter Persistent Homology to the Appendix. These sections are referred to and linked when required in the main text.
>
> We hope that these improvements will be helpful to the reviewers. We will update the reviewers promptly with any additional updates as they become available during the revision period.
>
> Below, we will respond to the reviewers' comments individually.

---

> > ### Author Response · Authors · 2025-11-28
> > **Release of Anonymised Code To Reproduce Numerical Experiments**
> >
> > We have updated the Reproducibility Statement and now provide an anonymous code repository with a Python implementation of MCbiF and scripts to reproduce all numerical experiments: https://anonymous.4open.science/r/mcbif-iclr-468A/.
> >
> > Making the code open access strengthens our work by facilitating reproducibility and enabling other researchers to build on MCbiF.

---

### Author Response · Authors · 2025-12-03
**Update on Numerical Experiments**

In our final update to our manuscript, we have provided additional numerical results on representation learning with graph convolutional networks (GCN) for the regression task and conducted a performance variability analysis for both regression and classification tasks. We summarise these additional improvements in the following:

- **Representation learning with GCNs:** Reviewer NmJi suggested benchmarking our method against additional representation learning methods in the regression task (Section 5.1) and suggested GCNs. Since our regression target is the minimal crossing number computed from the Sankey diagram layout, benchmarking against a GCN trained on the Sankey graphs is a natural baseline. We performed extensive hyperparameter optimisation for the GCN using the train and validation split of our data and evaluated the final performance on the test split (see Appendix D.1). We added our results to Tables 1 and 4 and found that the GCN performs better than the other representation learning methods (LR, MLP, CNN) trained on the raw label embeddings but **significantly worse** than models trained on the MCbiF feature maps. This complements our initial results on representation learning on the (non-unique) raw label encodings of $\theta$ with MLP and CNN, which we had performed previously. While GCN on Sankey graphs leads to higher performance than MLP and CNN on the raw label encodings, as expected, all representation learning models are still outperformed by models trained on our MCbiF feature maps by a large margin. We confirmed this through our performance variability analysis (see comment below) and statistical tests on the test residuals.

- **Performance variability analysis:** To measure the performance variability of the different methods in our regression and classification tasks as requested by reviewer NmJi, we computed bootstrapped 95% confidence intervals (CI) of the test $R^2$ score and test accuracy. We present our results in Tables 3 and 5 in the Appendix. Our analysis shows that MCbiF consistently outperforms the other methods. In particular, for the regression task the best MCbiF model (trained on combined HF0 and HF1 features) leads to a CI whose lower bound on the test $R^2$ score is higher than the upper bound of the CIs of the other methods. We also find the same consistency in the improved performance of HF1 features for the classification task.

During the discussion phase we have added substantially to the numerical experiments in our work by: (i) benchmarking against additional baseline methods for comparing partitions;  (ii) benchmarking against representation learning; and (iii) conducting a full performance variability analysis. All these experiments have been added to the main text (in expanded Tables 1 and 2) as well as in the Appendices (expanded Tables 3,4 and 5; expanded Fig 12) with some updated numbers due to extended and improved numerical runs. Our results confirm the superior performance of MCbiF topological features on downstream machine learning tasks for non-hierarchical sequences of partitions.

We also remark that, for full reproducibility, code for our method and all numerical experiments (including our latest additions) are available in our anonymous repository: https://anonymous.4open.science/r/mcbif-iclr-468A/

In addition to these extended numerical results, we have carried out an extensive update of our theoretical results, as detailed in our first responses to the reviewers. This has  improved and clarified the presentation of the theory, as well as providing further extensions of some of the results and making further connections clarifying how our framework extends other concepts in the literature.

We believe that we have thus fully addressed the comments by our reviewers and hope that our manuscript can now be accepted as a conference paper at ICLR.

---

### Meta-Review · Area_Chair_qENA · 2026-01-07

**Summary:**

The article has very positive reviews, except for NMji. The authors needed to address a few weaknesses, but they have not done so effectively. In fact, the response to NMji seems so unrelated to the question that I had to read it multiple times. For example, where do the authors respond to the use of synthetic datasets, and where do they explain why they do not add other real datasets?

Nonetheless, I base my overall positive recommendation on the reviewers' positivity. This article seems to be novel and has a nice theoretical background, with a "nice and elegant" filtration construction.

**Reviewer Concerns:**

NMji
- Evaluation does not contain topological embeddings or graph neural networks. The authors respond but do not address this question. It is unclear whether there are existing embeddings or GNNs for such datasets. Hence, authors may not be able to add a new dataset.
- Experiments are mostly on synthetic datasets. However, the reviewer does not suggest a new dataset.
- Code is not shared. The authors released the code.

A general review for that is this: The authors have shared the code, but they do not give new results for the GNN/embedding review for NMij, and given the synthetic data review, I do not think NMji would increase its score beyond a 4.

tfAn
- praises the article and does not really report a weakness.

5d9W
- points out small inconsistencies and a lack of related work, but in general reports no serious weakness.

**Reviewer Scores:**

The reviewers rated 2, 6 and 8. NmIj would increase 2 to 4, but others would keep their 6 and 8 scores.

---

### Decision · Program_Chairs · 2026-01-26

Accept (Poster)